# Nuclear NAD+-biosynthetic enzyme NMNAT1 facilitates development and early survival of retinal neurons

**David Sokolov[1], Emily R Sechrest[1], Yekai Wang[1,2], Connor Nevin[1], Jianhai Du[1,2], Saravanan Kolandaivelu[1,2]***

[1]Department of Ophthalmology and Visual Sciences, Eye Institute, One Medical Center Drive, West Virginia University, Morgantown, United States; [2]Department of Biochemistry, One Medical Center Drive, West Virginia University, Morgantown, United States

**Abstract** Despite mounting evidence that the mammalian retina is exceptionally reliant on proper NAD+ homeostasis for health and function, the specific roles of subcellular NAD+ pools in retinal development, maintenance, and disease remain obscure. Here, we show that deletion of the nuclear-localized NAD+ synthase nicotinamide mononucleotide adenylyltransferase-1 (NMNAT1) in the developing murine retina causes early and severe degeneration of photoreceptors and select inner retinal neurons via multiple distinct cell death pathways. This severe phenotype is associated with disruptions to retinal central carbon metabolism, purine nucleotide synthesis, and amino acid pathways. Furthermore, transcriptomic and immunostaining approaches reveal dysregulation of a collection of photoreceptor and synapse-specific genes in NMNAT1 knockout retinas prior to detectable morphological or metabolic alterations. Collectively, our study reveals previously unrecognized complexity in NMNAT1-associated retinal degeneration and suggests a yet-undescribed role for NMNAT1 in gene regulation during photoreceptor terminal differentiation.

***For correspondence:**
kolandaivelus@hsc.wvu.edu

**Competing interest:** The authors declare that no competing interests exist.

## Editor's evaluation

Mutations in the gene encoding the NMNAT1 enzyme cause Leber congenital amaurosis type 9 (LCA9), a blinding disease. Using conditional inactivation of the mouse gene in the retina, this study extends previous observations on the requirement of this enzyme for photoreceptors homeostasis. The study not only shows NMNAT1 is involved in photoreceptor terminal differentiation but also provides evidence that the survival of other cell types depends on this enzyme. It also shows that NMNAT1 deficiency leads to the activation of different cell death pathways and causes metabolic defects in retinal cells. The study thus provides a better picture of the retinal defects that may underlie LCA9 in humans.

## Introduction

Nicotinamide adenine dinucleotide (NAD+) is a ubiquitous cellular metabolite with a diverse palette of biological functions across all kingdoms of life. In addition to serving a central role in redox metabolism as an electron shuttle, NAD+ has well-defined roles as a substrate for a host of enzymes including sirtuins (SIRTs), mono- and poly-ADP-ribose polymerases (PARPs), and NAD+ glycohydrolases (CD38, CD157, and SARM1). Collectively, these roles implicate NAD+ metabolism in phenomena as diverse as aging, cell proliferation, immunity, neurodegeneration, differentiation, and development (*Houtkooper et al., 2010*; *Cantó et al., 2015*; *Nikiforov et al., 2015*;

*Cambronne and Kraus, 2020*; *Navas and Carnero, 2021*). A relatively recent advance in the field is the notion of compartmentalized NAD$^+$ metabolism—that regulation of NAD$^+$ in distinct subcellular compartments dictates function in diverse manners (*Cantó et al., 2015*; *Nikiforov et al., 2015*; *Cambronne and Kraus, 2020*; *Navas and Carnero, 2021*). While many aspects of this compartmentalization remain to be explored, it is now known that spatiotemporal NAD$^+$ regulation plays prominent roles in processes including axon degeneration, circadian regulation, and adipogenesis (*Cambronne and Kraus, 2020*).

Among mammalian tissues, the retina appears particularly reliant on proper NAD$^+$ homeostasis for survival and function. This is suggested by associations between retinal NAD$^+$ deficiency and pathology in diverse models of retinal dysfunction (*Lin et al., 2016*; *Williams et al., 2017*) as well as multiple mutations to NAD$^+$- or NADP$^+$-utilizing enzymes which cause blindness in humans (*Bowne et al., 2006*; *Aleman et al., 2018*; *Bennett et al., 2020*). Among these enzymes is nicotinamide mononucleotide adenylyltransferase-1 (NMNAT1), a highly conserved, nuclear-localized protein which catalyzes the adenylation of nicotinamide mononucleotide (NMN) or nicotinic acid mononucleotide (NaMN) to form NAD$^+$, the convergent step of all mammalian NAD$^+$ biosynthetic pathways (*Nikiforov et al., 2015*). To date, over 30 NMNAT1 mutations have been linked to the severe blinding diseases Leber congenital amaurosis type 9 (LCA9) and related cone-rod dystrophy (*Perrault et al., 2012*; *Falk et al., 2012*; *Chiang et al., 2012*; *Koenekoop et al., 2012*; *Coppieters et al., 2015*; *Nash et al., 2018*). Although NMNAT1 is ubiquitously expressed, and many of these mutations reduce NMNAT1 catalytic activity or stress-associated stability (*Falk et al., 2012*; *Koenekoop et al., 2012*; *Sasaki et al., 2015*), patients with these disorders rarely report extra-ocular phenotypes, a puzzling observation which is recapitulated by two LCA-NMNAT1 mutant mouse models (*Greenwald et al., 2016*). Further puzzling is the existence of two other NMNAT paralogs (Golgi-associated NMNAT2 and mitochondrial NMNAT3), which are detectable in the retina (*Kuribayashi et al., 2018*) but have not been linked to blindness. Importantly, while a crucial role for retinal NAD$^+$ was recently described through characterization of mice conditionally lacking the NAD$^+$ pathway enzyme NAMPT in photoreceptors (*Lin et al., 2016*), the significance of nuclear-synthesized NAD$^+$ in vision—suggested by the fact that NMNAT1 is the only NAD$^+$-pathway enzyme to date linked to blindness—remains poorly understood.

Current results point to multiple, potentially distinct roles for NMNAT1 in the retina—ex vivo studies suggest that NMNAT1 supports sirtuin function to facilitate the survival of retinal progenitor cells (*Kuribayashi et al., 2018*), while ablation of NMNAT1 in mature mice results in rapid death of photoreceptors mediated by the neurodegenerative NADase SARM1 (*Sasaki et al., 2020b*). Global deletion of NMNAT1 in mice is embryonically lethal (*Conforti et al., 2011*), suggesting non-redundant roles for nuclear NAD$^+$ synthesis during development. Consistent with this notion, pan-retinal NMNAT1 deletion is shown to cause rapid and severe retinal degeneration in mice shortly after birth (*Wang et al., 2017*; *Eblimit et al., 2018*). While these studies suggest diverse functions of retinal NMNAT1 beyond its canonical role in redox metabolism, the degree to which these functions overlap—as well as the mechanistic basis for the severity of NMNAT1-associated retinal dystrophy in animal models and patients—have not been comprehensively explored.

In this study, we investigate the roles of NMNAT1-mediated NAD$^+$ metabolism in the retina by generating and characterizing a retina-specific NMNAT1 knockout mouse model. Utilizing histological and transcriptomic approaches, we demonstrate that NMNAT1 deletion causes severe and progressive retinal degeneration affecting specific retinal cell types beyond photoreceptors, and that this severe degeneration likely results from activation of multiple distinct cell death pathways. Comprehensive metabolomics analysis reveals specific metabolic defects in NMNAT1 knockout retinas and suggests impaired central carbon, purine nucleotide, and amino acid metabolism as a cause for severe degeneration. Strikingly, RNA-sequencing reveals a collection of photoreceptor and synapse-specific genes which are downregulated in knockout retinas preceding degeneration. Immunostaining of several of these genes suggests severe impairment of photoreceptor terminal differentiation in the absence of NMNAT1. Overall, our results reveal a previously unappreciated complexity in NMNAT1-associated retinal degeneration, provide possible explanations for the retina-specific manifestations of NMNAT1 deficiency, and propose a yet-undescribed role for NMNAT1 in gene regulation during late-stage retinal development.

## Results

### Generation and validation of NMNAT1 conditional knockout mouse model

To establish a retina-specific NMNAT1 knockout model, we crossed mice homozygous for a loxP-targeted *Nmnat1* locus (*Nmnat1*<sup>flox/flox</sup>) with transgenic mice expressing Cre recombinase under a *Six3* promoter (*Nmnat1*<sup>wt/wt</sup>;*Six3-Cre*), which is expressed throughout the retina at embryonic day 9.5 (E9.5) and shows robust activity by E12.5 (*Furuta et al., 2000*). After several crosses, mice inheriting *Six3-Cre* and a floxed *Nmnat1* locus (*Nmnat1*<sup>flox/flox</sup>;*Six3-Cre*, hereafter referred to as 'knockouts') exhibit Cre-mediated excision of the first two exons of *Nmnat1*—which contain important substrate binding domains—in the embryonic retina (*Figure 1A*). We determined that retinal *Nmnat1* expression in postnatal day 4 (P4) knockout mice was reduced by 75.6% (95% CI 56.1–95.0%) compared to littermate controls (*Figure 1B*), a reduction consistent with a previously reported NMNAT1 retinal knockout model (*Sasaki et al., 2020b*). We further verified that retinal NMNAT1 protein levels were drastically reduced in P0 knockout mice using a custom-made polyclonal antibody against NMNAT1 (*Figure 1C* and *Figure 1—figure supplement 1*). Finally, we confirmed that embryonic *Six3-Cre* expression alone does not cause gross retinal abnormalities by staining for several well-characterized cell type markers in mature *Nmnat1*<sup>wt/wt</sup>; *Six3-Cre* retinas and littermate controls (*Figure 2—figure supplement 1*; markers discussed below).

### Early-onset and severe morphological defects in the NMNAT1-null retina

As a first step toward characterizing the effects of NMNAT1 ablation on the retina, we performed retinal histology using hematoxylin and eosin (H&E) staining. H&E-stained retinal cross sections from P0 knockout and control mice reveal no obvious morphological differences (*Figure 1D and H*); however, by P4, knockout retina are markedly thinner than controls and exhibit disrupted lamination and evidence of large-scale cell death in the inner and outer nuclear layers (*Figure 1E*). Degeneration is most severe in the central retina, with a ~45% reduction in central retinal thickness (but unaffected peripheral retinal thickness) in P4 knockout mice (*Figure 1I*). By P10, knockouts show a ~62% reduction in central retinal thickness and ~27% reduction in peripheral retinal thickness compared to controls (*Figure 1J*). Degeneration of the entire inner and outer nuclear layers is nearly complete by P30 (*Figure 1G*), while remaining inner retinal structures persist until approximately P60 (data not shown). Proper segregation of inner and outer retinal neurons appears disrupted in P4 knockouts, but this segregation is established in P10 knockouts despite severe degeneration (*Figure 1F*). Interestingly, formation of the outer plexiform layer (containing photoreceptor and bipolar neuron synaptic structures) appears disrupted in P4 and P10 knockout retinas (*Figure 1E and F*).

### NMNAT1 loss affects survival of major inner retinal neurons

Histological examination suggests severe photoreceptor degeneration in NMNAT1 knockout retinas but also indicates loss of specific inner retinal neuron populations. To further characterize these effects, we quantified populations of several major inner retinal cell types in our knockout by staining retinal sections with well-characterized antibody markers: retinal ganglion cells were identified by labelling for brain-specific homeobox/POU domain protein 3A (BRN3A), amacrine cells by labeling for calretinin (CALR), horizontal cells by labeling for calbindin (CALB), and bipolar cells by labeling for Ceh-10 homeodomain-containing homolog (CHX10) (**Key Resources Table**). With the exception of calbindin, we performed this analysis at P4 and P10—representing early and late stages of degeneration, respectively—revealing an interesting cell-type-dependent sensitivity to NMNAT1 loss (*Figure 2*). At both tested ages, relative numbers of retinal ganglion cells are not significantly different between knockout and control retinas (*Figure 2A–D and M*), while numbers of amacrine cells are unchanged at P4 but reduced by ~51% (95% CI 36–65%) in P10 knockout retinas (*Figure 2E–H and O*). Numbers of bipolar cells are similarly unchanged at P4 but reduced by ~75% (95% CI 62–85%) in P10 knockout retinas (*Figure 2I–L and P*). In P0 knockout retinas—an age at which ganglion, amacrine, and bipolar cell numbers are unchanged (data not shown)—horizontal cell counts are reduced by ~36% (95% CI 18–53%) (*Figure 2N*), and this trend persists in P4 knockout retinas (*Figure 2A–B and N*). These results identify retinal bipolar, horizontal, and amacrine neurons as targets of NMNAT1-associated

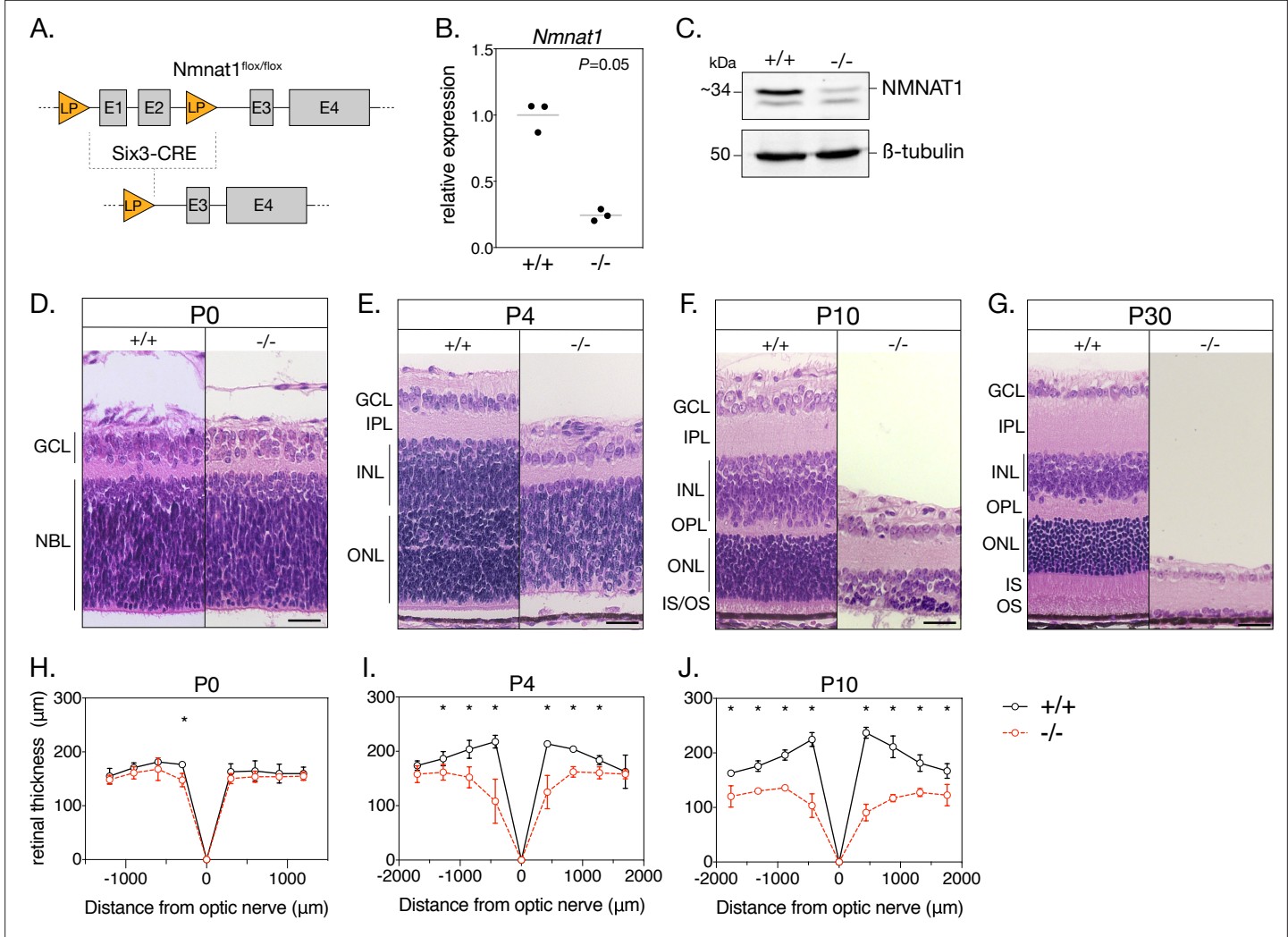

**Figure 1.** Loss of NMNAT1 leads to early and severe retinal degeneration. (**A**) Schematic depicting retina-specific *Six3-Cre* mediated excision of a segment of the *Nmnat1* gene. (**B**) Relative *Nmnat1* expression in retina from P4 knockout (-/-) and littermate control (+/+) mice as assessed by RT-qPCR (grey bars represent mean, significance determined using Mann-Whitney U test, n = 3 biological replicates). (**C**) Representative western blot showing levels of NMNAT1 and β-tubulin loading control in retinal lysate from P0 knockout and control mice. (**D–G**) Representative H&E-stained retinal cross sections from knockout and control mice at indicated ages. (**H–J**) Spider plots depicting mean retinal thickness at P0, P4, and P10. Data are represented as mean ± SD. *p < 0.05 using Student's t-test, n = 3 biological replicates per age. Scale bars, 30 μm. Abbreviations: LP, loxP site; E1-4, exon 1–4; P, postnatal day; GCL, ganglion cell layer; NBL, neuroblastic layer; IPL, inner plexiform layer; OPL, outer plexiform layer; INL, inner nuclear layer; ONL, outer nuclear layer; IS/OS, photoreceptor inner segment/outer segment layer.

The online version of this article includes the following source data and figure supplement(s) for figure 1:

**Source data 1.** Quantification of *Nmnat1* mRNA levels in P0 WT and KO retinas.

**Figure supplement 1.** Validation of anti-NMNAT1 antibody in cell lines and retinal tissue.

**Figure supplement 1—source data 1.** Raw western blot scan of P4 WT and KO retinal lysate stained with anti-NMNAT1 antibody (700 channel).

**Figure supplement 1—source data 2.** Raw western blot scan of P4 WT and KO retinal lysate stained with anti-beta-tubulin antibody (800 channel).

**Figure supplement 1—source data 3.** Raw western blot scan of HEK293T cell lysate stained with anti-NMNAT1 antibody (700 channel).

**Figure supplement 1—source data 4.** Raw western blot scan of HEK293T cell lysate stained with anti-FLAG antibody (800 channel).

**Figure supplement 1—source data 5.** Uncropped western blot scan of P4 WT and KO retinal lysate stained with anti-NMNAT1 antibody.

**Figure supplement 1—source data 6.** Uncropped western blot scan of P4 WT and KO retinal lysate stained with anti-beta-tubulin antibody.

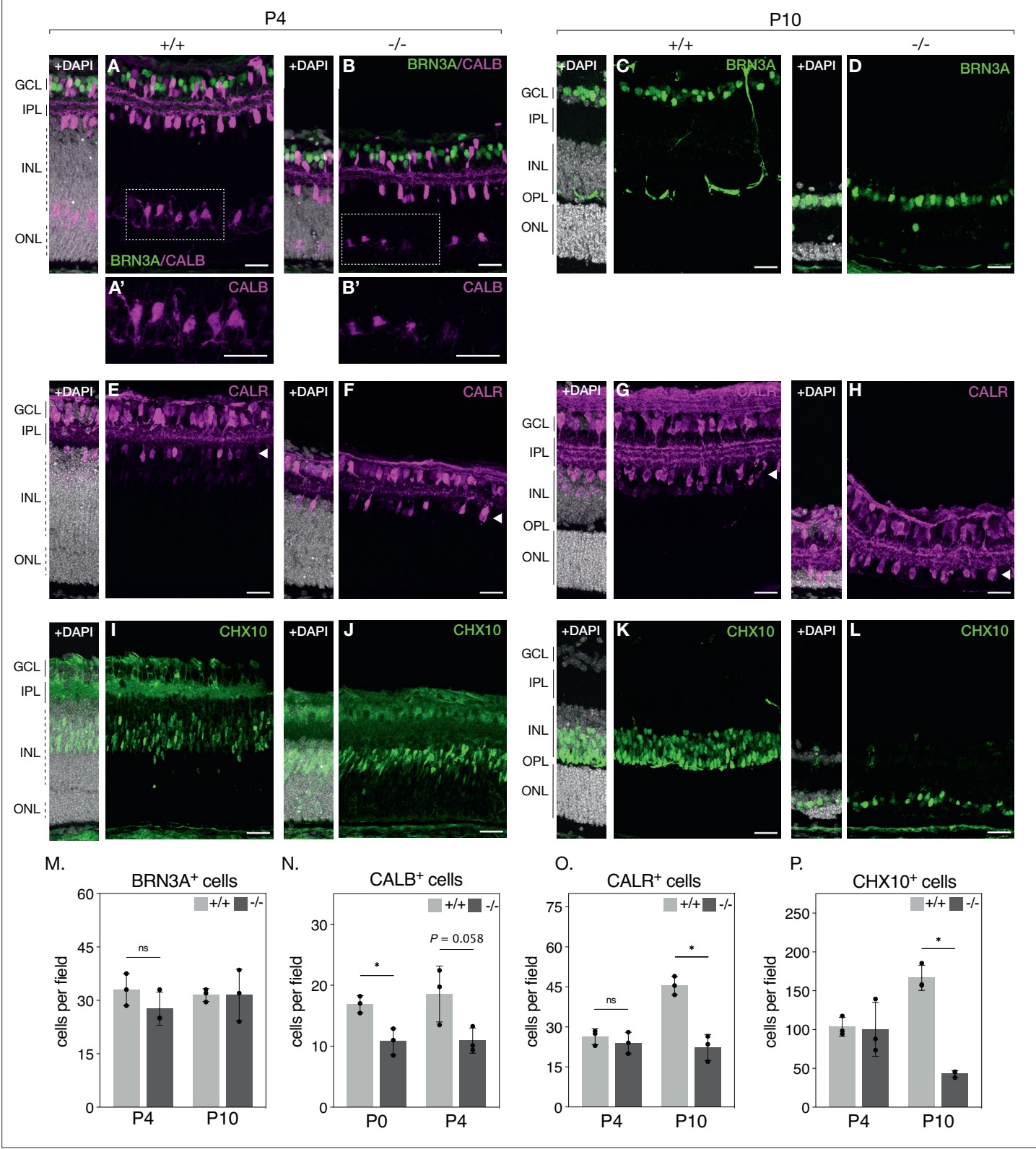

**Figure 2.** NMNAT1 loss affects retinal bipolar, horizontal and amacrine cells. Representative retinal sections from knockout (-/-) and floxed littermate control (+/+) mice at the indicated ages labeled with antibodies against BRN3A (**A-D, green**), Calbindin (CALB) (**A–B**, magenta) Calretinin (CALR) (**E–H**), and CHX10 (**I–L**). Quantification of BRN3A (**M**), CALB, (**N**), CALR (**O**), and CHX10-positive cells (**P**) are shown. In (**O**), only CALR-positive cells on the outer side of the IPL (layer indicated by white arrowheads) were counted. Data is represented as mean ± SD. n = 3 biological replicates for all panels;

*Figure 2 continued on next page*

*Figure 2 continued*

significance determined using Student's t-test. Scale bars, 30 μm.

The online version of this article includes the following source data and figure supplement(s) for figure 2:

**Source data 1.** Numerical source data for retinal cell type quantification in P4 and P10 KO and WT retinas.

**Figure supplement 1.** Six3-Cre does not cause obvious defects in the mature retina.

pathology and suggest that horizontal and bipolar neurons are more sensitive to NMNAT1 loss than amacrine neurons. Interestingly, while retinal ganglion cells do eventually degenerate at timepoints past P30 (data not shown), they appear largely agnostic to NMNAT1 loss in the young postnatal retina.

## Loss of NMNAT1 impairs photoreceptor terminal differentiation

Turning our attention to photoreceptors, we repeated the above approach with antibodies against the photoreceptor markers recoverin (anti-RCVRN) and rhodopsin (anti-RHO). While anti-RCVRN cleanly labels developing photoreceptor somas in P4 and P10 control retinas (***Figure 3H***, ***Figure 3—figure supplement 1F***), we observe a complete lack of recoverin expression in knockout retinas at both ages (***Figure 3I***, ***Figure 3—figure supplement 1G***). Barring a small amount of non-specific staining likely originating from the secondary antibody (***Figure 3—figure supplement 1D,E***), rhodopsin expression at P4 and P10 showed an identical trend to that of recoverin (***Figure 3J and K***, ***Figure 3—figure supplement 1H,I***).

Intrigued by the magnitude of recoverin and rhodopsin loss and hypothesizing defects in the expression of other retinal proteins in our knockout, we comprehensively profiled the transcriptomes of knockout and control retinas at two timepoints— pre-degeneration (E18.5) and during degeneration (P4) and using RNA-sequencing. At P4, this analysis reveals 2976 differentially expressed genes in NMNAT1 knockout retinas (***Figure 3—figure supplement 2B***), several of which we validated using RT-qPCR (***Figure 3—figure supplement 1J***). Consistent with the lack of recoverin and rhodopsin staining at this age, gene set enrichment analysis (GSEA) of P4 differentially expressed genes (DEGs) reveals several large, highly-overrepresented clusters of downregulated photoreceptor-related genes including both recoverin and rhodopsin (***Figure 3—figure supplement 2C***). Strikingly, among 815 DEGs in E18.5 knockout retinas, a similar cluster of downregulated genes associated with visual perception and the photoreceptor outer segment was observed (***Figure 3—figure supplement 3B,C***). Combining both RNA-sequencing datasets reveals a group of 365 DEGs in knockout retinas common to both timepoints (***Figure 3L***). Importantly, GSEA on this gene set reveals highly overrepresented clusters of photoreceptor and synapse associated genes (***Figure 3M***), and further analysis identifies a core set of 21 photoreceptor-associated genes which are significantly downregulated in E18.5 and P4 NMNAT1 knockout retinas (***Figure 3N and O***). Notably, this set includes rod-specific (e.g. *Gngt1*), cone-specific (e.g. *Opn1sw*, *Cnga3*) and photoreceptor-specific (e.g. *Prph2*, *Rcvrn*, *Aipl1*) genes of diverse function, many of which have important roles in photoreceptor development and function. Consistent with a specific transcriptional effect on photoreceptors, we confirmed that expression of several well-known ganglion cell, amacrine/horizontal cell, and bipolar cell-specific genes was largely unchanged in NMNAT1 knockout retinas at either tested age (***Figure 3—figure supplement 4***).

To further confirm the relevance of our RNA-sequencing results, we immunostained knockout and control retinas with several markers of developing cone photoreceptors: anti-phosducin (PDC) (***Rodgers et al., 2016***), anti-M-opsin (OPN1MW), and peanut agglutinin (PNA), which labels developing cone outer segments (***Blanks and Johnson, 1984***). While these markers showed normal cone accumulation and rudimentary outer segment formation in the OPL of P0 and P4 control retinas (***Figure 3E and G***), P0 knockout retinas showed markedly reduced and widely dispersed phosducin-positive cones (***Figure 3F***), and P4 knockouts demonstrated a near complete absence of M-opsin expression, mirroring recoverin and rhodopsin at this age (***Figure 3H***). Accordingly, knockouts at both ages show significant attenuation of cone outer segment formation as determined by PNA staining (***Figure 3F and H***).

Due to the extent of photoreceptor-specific transcript and protein alterations in knockout retinas, we hypothesized that NMNAT1 is essential for photoreceptor terminal differentiation—the process by which partially committed retinal progenitors initiate photoreceptor-specific transcriptional and morphological programs (***Swaroop et al., 2010***; ***Brzezinski and Reh, 2015***; ***Daum et al., 2017***).

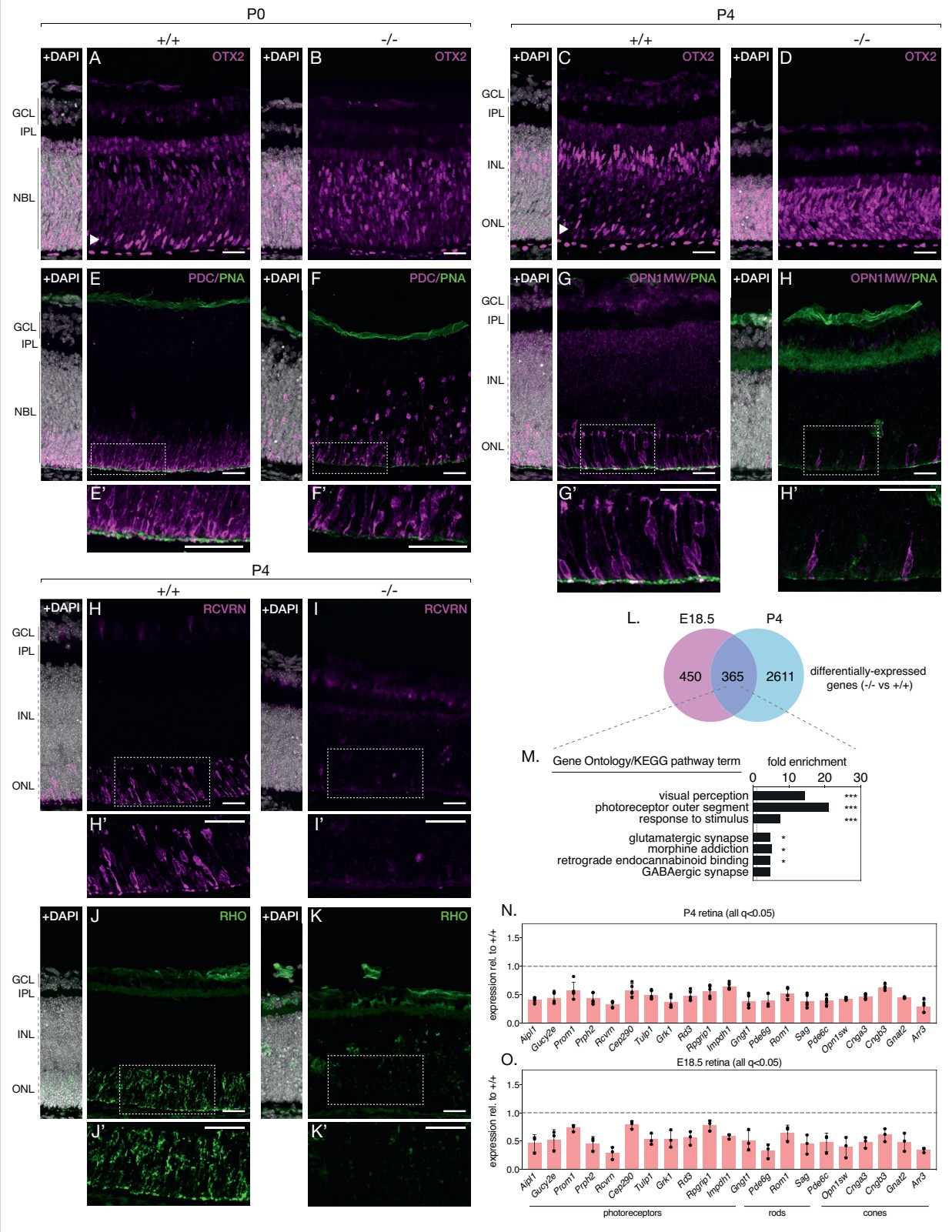

**Figure 3.** NMNAT1 loss impairs photoreceptor terminal differentiation. Representative retinal sections from knockout (-/-) and floxed littermate control (+/+) mice at the indicated ages labelled with antibodies against OTX2 (**A–D**), Phosducin (PDC) and peanut agglutinin (PNA) (**E, F**), M-opsin (OPN1MW) and PNA (**G, H**), recoverin (RCVRN) (**H,I**) or rhodopsin (RHO) (**J, K**). (**L**) Comparison of differentially expressed genes in E18.5 and P4 knockout retinas as assessed by RNA-sequencing. (**M**) GSEA of differentially expressed genes identified in E18.5 and P4 knockout retinas. (**N, O**) Relative expression of

*Figure 3 continued on next page*

*Figure 3 continued*

indicated genes in P4 and E18.5 knockout retinas as assessed by RNA-sequencing. n = 3 biological replicates for all panels, except (**N**), where n = 5 biological replicates. Corresponding zoom panels are indicated with dotted rectangles. Scale bars, 30 μm.

The online version of this article includes the following source data and figure supplement(s) for figure 3:

**Source data 1.** Gene ontology (GO) overrepresentation analysis of genes differentially expressed in KO retinas at E18.5 and P4 timepoints.

**Figure supplement 1.** Proliferation is only modestly affected in P0 NMNAT1 knockout retinas.

**Figure supplement 1—source data 1.** Numerical source data for expression (relative to WT) of select genes in P4 KO retinas using RNA-seq and RT-qPCR.

**Figure supplement 1—source data 2.** Numerical source data for quantification of PHH3-positive nuclei in P0 WT and KO retinal sections.

**Figure supplement 2.** Global transcriptional changes in P4 NMNAT1 knockout retinas.

**Figure supplement 2—source data 1.** Gene ontology (GO) overrepresentation analysis of significantly changed genes in P4 KO retinas.

**Figure supplement 3.** Global transcriptional changes in E18.5 NMNAT1 knockout retinas.

**Figure supplement 3—source data 1.** Gene ontology (GO) overrepresentation analysis of significantly changed genes in E18.5 KO retinas.

**Figure supplement 4.** Expression of a collection of non-photoreceptor specific genes is largely unchanged in NMNAT1 knockout retinas.

**Figure supplement 4—source data 1.** Numerical source data for expression (relative to WT) of retinal cell-type-specific genes in E18.5 and P4 KO retinas from RNA-seq dataset.

**Figure supplement 5.** NMNAT1-loss during retinal development affects formation of the outer plexiform layer.

**Figure supplement 5—source data 1.** Numerical source data for expression (relative to WT) of several synapse-specific genes in E18.5 and P4 KO retinas.

To test this possibility, we probed knockout and control retinas with an antibody against OTX2, a well-characterized transcription factor necessary for photoreceptor terminal differentiation (*Beby and Lamonerie, 2013*). In P0 and P4 control retinas, OTX2 is enriched in developing photoreceptors in the ONL (*Figure 3A and C*, arrowheads) and in developing bipolar neurons in the INL (*Figure 3C*). Consistent with our hypothesis, OTX-positive photoreceptors are absent in P0 knockout retinas (*Figure 3B*), and P4 knockout retinas display a striking expression of OTX2 throughout the ONL/INL (*Figure 3D*). Finally, to determine whether impaired retinal progenitor proliferation may explain reduced photoreceptor numbers in knockouts, we assessed proliferation at P0 using the well-characterized marker phospho-histone H3 (PHH3), which labels M-phase cells (*Prigent and Dimitrov, 2003*). This analysis revealed a slight but not significant decrease in PHH3-positive cells in P0 knockout retinas (*Figure 3—figure supplement 1A–C*) Altogether, these results argue that NMNAT1 is crucial for terminal differentiation—but not early proliferation—of retinal photoreceptors and suggest photoreceptor-specific transcriptional dysregulation as a driver of the severe photoreceptor phenotype in NMNAT1-deficient retinas.

Beyond affecting photoreceptor-specific gene expression, we also note downregulation of 6 synapse-associated genes (*Stx3*, *Syngr1*, *Cln3*, *Scamp5*, and *Sv2b*) in both E18.5 and P4 knockout retinas (*Figure 3—figure supplement 5C,D*), consistent with disruptions to outer plexiform layer formation in P4 knockout retinas on histology and on staining with the synapse marker synaptophysin (anti-SYPH) (*Figure 3—figure supplement 5*).

## Loss of NMNAT1 during retinal development triggers multiple cell death pathways

As NMNAT1 deficiency drastically impairs the postnatal survival of photoreceptor, bipolar, horizontal, and amacrine retinal neurons, we sought to determine the mechanisms by which these cells degenerate. To this aim, we began by staining retinal sections with an antibody against activated caspase-3 (AC3). While P0 knockout retinas show little AC3 staining compared to controls—consistent with grossly normal retinal morphology at this age—P4 knockout retinas show robust AC3 immunoreactivity in the inner and outer nuclear layers (*Figure 4A–D*). As expected, most AC3-immunoreactive (AC3⁺) cells display nuclear chromatin condensation ('pyknosis') characteristic of dying cells; however, staining also revealed a population of pyknotic nuclei not immunoreactive to AC3 (AC3⁻) (*Figure 4A'–D'*, **arrows**). Interestingly, these pyknotic, AC3⁻ nuclei were sparsely present in P0 and P4 control retinas (*Figure 4A and C*) and to a larger extent in P0 and P4 knockout retinas (*Figure 4B and D*). Quantification (*Figure 4E*) reveals a general trend of cell

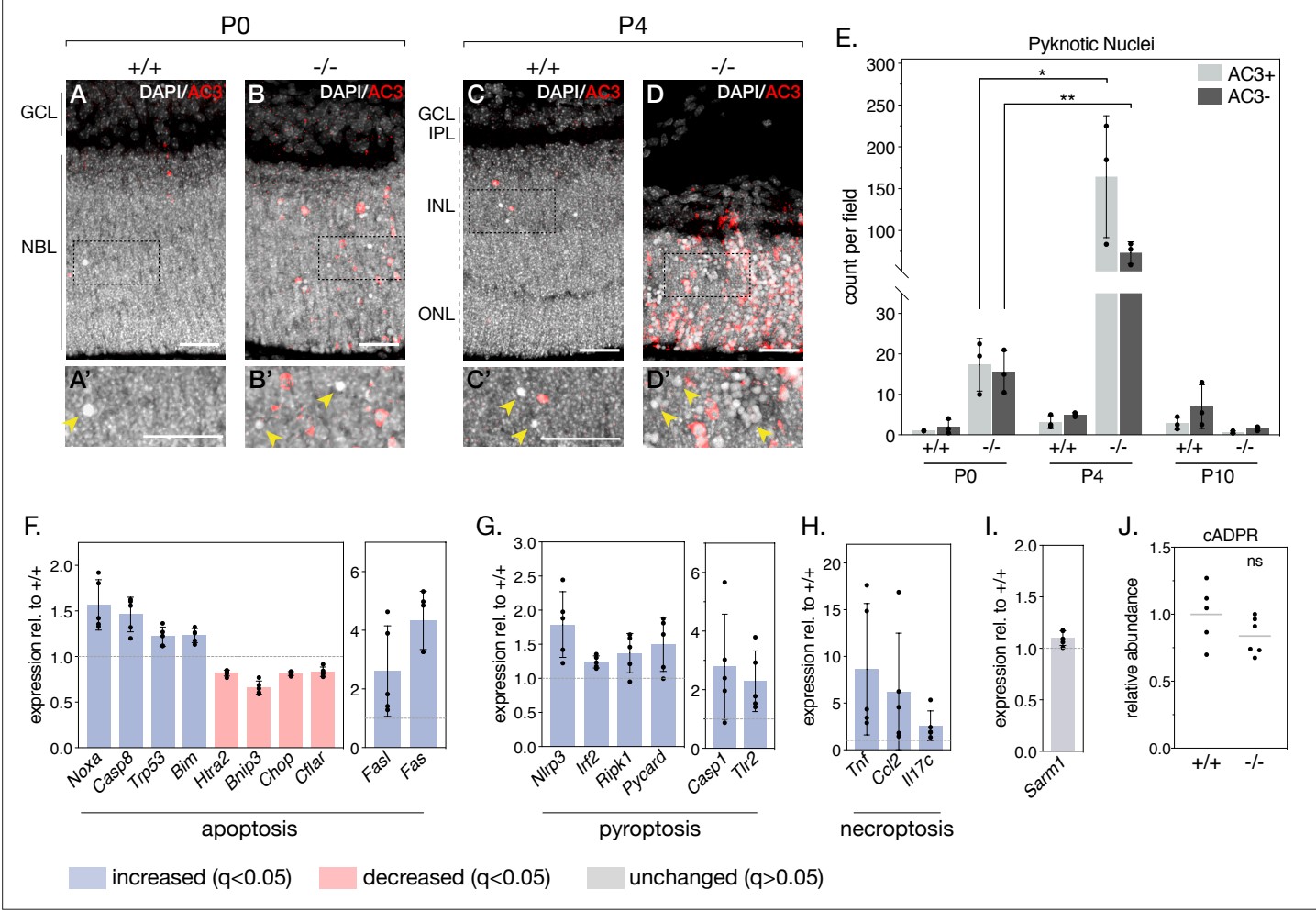

**Figure 4.** NMNAT1 loss causes activation of multiple cell death pathways in the retina. (**A–D**) Representative retinal sections from knockout (-/-) and floxed littermate control (+/+) mice at the indicated ages labelled with an antibody against active Caspase-3 (AC3). Corresponding zoom panels are indicated with dotted rectangles. Arrows denote pyknotic, AC3-negative nuclei. (**E**) Quantification of pyknotic nuclei in sections from knockout and control mice at the indicated ages, grouped by presence (AC3+) or absence (AC3-) of active Caspase-3 labeling. Relative expression of several apoptotic (**F**), pyroptotic (**G**), and necroptotic (**H**) genes in P4 knockout retinas as assessed by RNA-sequencing. (**I**) Relative expression of *Sarm1* in P4 knockout and control retinas as assessed by RNA-sequencing. (**J**) Relative abundance of cyclic-ADP-ribose (cADPR) in P4 knockout and control retinas as measured by mass spectrometry (grey bars represent means). Data are represented as mean ± SD. Significance determined using unpaired t-tests for (**E**) and (**J**) or DESeq2 for (**F–I**) (see Materials and methods). n = 3 biological replicates per condition for (**A–E**), n = 5 biological replicates for (**F–I**), n = 6 biological replicates (one outlier removed) for (**J**). Scale bars, 30 μm.

The online version of this article includes the following source data and figure supplement(s) for figure 4:

**Source data 1.** Numerical source data for quantification of Caspase-3-positive (AC3+) and Caspase-3-negative (AC3-) pyknotic cells in WT and KO retinal sections.

**Figure supplement 1.** Deregulation of several cell death pathways in NMNAT1-null retinas preceding gross degeneration.

**Figure supplement 1—source data 1.** Numerical source data for expression (relative to WT) of cell death-related genes in E18.5 KO retinas.

**Figure supplement 1—source data 2.** Uncropped western blot of P4 KO and WT retinal lysate stained with anti-Gasdermin D (GSDMD) antibody.

**Figure supplement 1—source data 3.** Uncropped western blot of P4 KO and WT retinal lysate stained with anti-beta tubulin antibody.

**Figure supplement 1—source data 4.** Raw western blot scan of P4 WT and KO retinal lysate stained with anti-Gasdermin D (GSDMD) antibody (700 channel).

**Figure supplement 1—source data 5.** Raw western blot scan of P4 WT and KO retinal lysate stained with anti-beta tubulin antibody (800 channel).

**Figure supplement 2.** Transcriptional survey of cell death in the NMNAT1-null retina.

**Figure supplement 2—source data 1.** Numerical source data for expression of cell death pathway genes in P4 WT and KO retinas.

death consistent with our histology and cell-marker investigations: nuclear pyknosis is slightly elevated in P0 knockout retinas compared to controls, peaks at P4 where we observe robust retinal degeneration, and is virtually absent by P10, by which time the majority of outer and inner nuclei in the knockout are lost (*Figure 1F*). Interestingly, we observe roughly equal amounts of AC3$^+$ and AC3$^-$ pyknotic cells in P0 knockout retinas, whereas by P4 AC3$^-$ pyknotic cells constitute ~30% of pyknotic cells in knockout retinas (*Figure 4E*). In addition to being present at all tested ages and following the same general trend as AC3+ pyknotic cells, AC3$^-$ pyknotic cells often appear in distinct clusters (*Figure 4D'*), distinguishable from the more evenly dispersed AC3$^+$ pyknotic cells. These results suggest the activation of at least two distinct cell death pathways in NMNAT1 knockout retinas between P0 and P4.

To more comprehensively characterize NMNAT1-associated cell death and identify possible caspase 3-independent cell death pathways in our knockout, we leveraged the E18.5 and P4 RNA-sequencing datasets mentioned above. This allowed us to systematically assay the expression of a collection of genes associated with several major cell death pathways (*Figure 4—figure supplement 2*). Consistent with AC3 staining, we observe deregulation of a collection of apoptosis-related genes in P4 knockout retina, including significant increases in *Noxa* and *Fas*, two pro-apoptotic genes previously associated with cell death in NMNAT1-deficient retinas (*Kuribayashi et al., 2018*; *Figure 4F*). Notably, two of these genes—*Noxa* and *Chop*—are also significantly deregulated at E18.5, prior to significant retinal degeneration (*Figure 4—figure supplement 1A*).

In addition to transcriptional signatures of apoptosis, we identified upregulation of a collection of genes associated with pyroptosis in P4 NMNAT1 knockout retinas (*Figure 4G*). Pyroptosis is characterized by assembly of a multi-protein complex called the 'inflammasome,' which ultimately cleaves and activates the pore-forming members of the gasdermin family of proteins to elicit lytic cell death in response to a variety of perturbations (*Swanson et al., 2019*; *McKenzie et al., 2020*). Interestingly, we find upregulation of all three classical inflammasome components—*Nlrp3*, *Casp1*, and *Pycard* (ASC)—in P4 knockout retinas (*Figure 4G*), with *Nlrp3* upregulation at E18.5 as well (*Figure 4—figure supplement 1B*). In addition, we observe significant increases in *Irf2*, a transcriptional activator of gasdermin D (*Kayagaki et al., 2019*), as well as pyroptosis-associated proteins *Ripk1* and *Tlr2* at P4 (*Figure 4G*). Notably, expression of *Tlr2* and related protein *Tlr4* is also significantly elevated in E18.5 knockout retinas (*Figure 4—figure supplement 1B*). Finally, we also observed dysregulation of several genes associated with necroptosis (*Figure 4H*) and ferroptosis (*Figure 4—figure supplement 2D*) in P4 knockout retinas; while none of these genes were significantly upregulated at E18.5, we do observe an early induction of necroptosis-associated protein *Nox2* at this age (*Figure 4—figure supplement 1C*).

Recently, photoreceptor cell death in a postnatally induced global NMNAT1 knockout mouse was shown to depend heavily on the activity of the pro-degenerative axonal protein SARM1 (*Sasaki et al., 2020b*). Reasoning SARM1 as the culprit behind the caspase 3-independent cell death in our model, we checked *Sarm1* expression in our RNA-seq data and assayed SARM1 activity by measuring levels of its catalytic product cyclic ADP-ribose (cADPR) using targeted mass spectrometry in P4 and E18.5 NMNAT1 knockout and control retinas. Surprisingly, we found no significant changes in SARM1 expression (*Figure 4I*, *Figure 4—figure supplement 1D*) or activity (*Figure 4J*, *Figure 4—figure supplement 1E*) at either tested age. Overall, these data reveal that activation of multiple cell death pathways underlies the early and severe degeneration observed in NMNAT1 knockout retinas, and suggest pyroptosis, necroptosis and apoptosis as potential drivers of this degeneration.

## Global metabolic alterations in NMNAT1-deficient retinas

To identify possible mechanisms for the severe and cell-type-specific retinal degeneration in our model, we next sought to characterize global metabolic consequences of embryonic NMNAT1 deletion in the retina. To this end, we used targeted liquid chromatography-tandem mass spectrometry (LC-MS/MS) to quantify levels of ~112 cellular metabolites spanning many essential biochemical pathways in NMNAT1 knockout and control retinas at pre- and post-degenerative timepoints matching that of our RNA-sequencing analyses (E18.5 and P4). While LC-MS/MS analysis revealed no significant changes in E18.5 knockout retinas compared to controls, analysis at P4 revealed significantly altered levels of 39 metabolites in knockout retinas (*Figure 5*, *Figure 5—figure supplement 1*). Metabolite

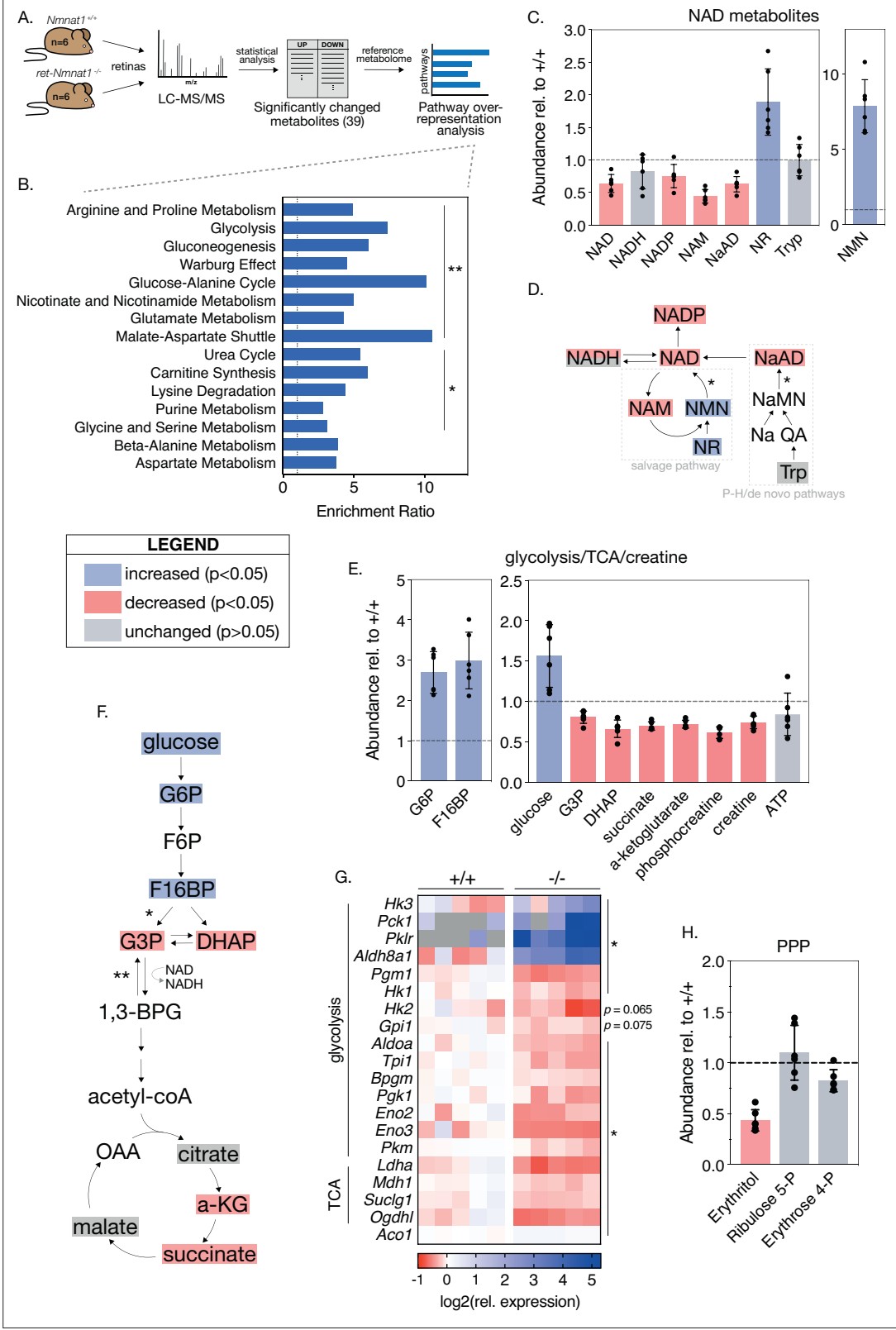

**Figure 5.** Loss of NMNAT1 impairs retinal central carbon metabolism. (**A**) Schematic illustrating metabolomics experimental approach. (**B**) Results of metabolite set enrichment analysis (MSEA) on significantly changed metabolites in P4 knockout retinas. (**C**) Relative abundance of NAD⁺ pathway metabolites in P4 knockout retinas as assessed by mass spectrometry. (**D**) Schematic illustrating the major mammalian NAD⁺ synthesis pathways

*Figure 5 continued on next page*

*Figure 5 continued*

colored according to metabolite changes in (**C**); NMNAT1-catalyzed steps are indicated with asterisks. (**E**) Relative abundance of glycolysis, TCA cycle, and creatine metabolites in P4 knockout retinas as assessed by mass spectrometry. (**F**) Schematic depicting abbreviated glycolysis/TCA cycle pathway colored according to metabolite changes in (**E**); aldolase-catalyzed step is indicated by a single asterisk, while GAPDH-catalyzed step is marked with a double asterisk. (**G**) Heatmap of log-transformed relative expression of a set of glycolysis/TCA cycle genes in P4 knockout (-/-) and control retinas (+/+) as assessed by RNA-sequencing. (**H**) Relative abundance of pentose-phosphate pathway (PPP) metabolites in P4 knockout retinas as assessed by mass spectrometry. Data are represented as mean ± SD. n = 6 biological replicates for (**B,C,E,H**), n = 5 biological replicates for (**G**). Statistical significance in panels (**C**), (**E**), and (**H**) is denoted according to the boxed legend.

The online version of this article includes the following source data and figure supplement(s) for figure 5:

**Source data 1.** Metabolic pathway overrepresentation analysis on significantly changed metabolites in P4 KO retinas.

**Source data 2.** Numerical source data for E18.5 and P4 LC-MS/MS experiments; peak intensity tables for all detected metabolites in E18.5 and P4 KO and WT retinas.

**Figure supplement 1.** Additional metabolic changes in NMNAT1-null retinas.

**Figure supplement 1—source data 1.** Numerical source data for abundance (relative to WT) of acylcarnitine species in P4 KO retinas.

**Figure supplement 1—source data 2.** Numerical source data for abundance (relative to WT) of nucleotide and amino acid metabolites in P4 KO retinas.

**Figure supplement 2.** NMNAT1-associated changes in retinal nucleotide and amino acid metabolism.

set enrichment analysis (MSEA) of altered metabolites identifies potential disruption of several diverse biochemical pathways including amino acid metabolism, glycolysis/gluconeogenesis, nicotinate and nicotinamide metabolism, and purine metabolism (*Figure 5B*).

NMNAT1 knockout retinas show specific metabolic disruptions to NAD$^+$ biosynthesis pathways. At P4, knockouts show a ~40% reduction of total retinal NAD$^+$ levels and levels of nicotinic acid adenine dinucleotide (NaAD), the other catalytic product of NMNAT1 (*Figure 5C and D*). Levels of the downstream metabolites NADP and nicotinamide (NAM) were decreased by ~25% and ~55%, respectively, while levels of NADH were slightly decreased (p > 0.05) (*Figure 5C*). As expected, we observed significant accumulation of NAD$^+$ precursors nicotinamide riboside (NR) and nicotinamide mononucleotide (NMN) in P4 knockout retinas (*Figure 5C*); however, we observed no significant changes in levels of tryptophan, the starting point for de novo NAD$^+$ synthesis, at this age (*Figure 5C and D*).

## Retinal NMNAT1 loss causes disruption of central carbon metabolism

Interestingly, the set of significantly altered metabolites in P4 knockout retinas is enriched for metabolites associated with glycolysis, gluconeogenesis, and the Warburg effect (*Figure 5B*). Closer examination of these pathways reveals large relative increases in levels of the upstream glycolytic metabolites glucose, glucose 6-phosphate (G6P) and fructose 1,6-bisphosphate (F16BP), as well as significant decreases in levels of dihydroxyacetone phosphate (DHAP) and glucose 3-phosphate (G3P) (*Figure 5E and F*), strongly suggesting a disruption to glucose utilization. Consistent with such an effect, levels of the TCA cycle intermediates alpha-ketoglutarate (a-KG) and succinate are decreased by ~30% in P4 knockout retinas (*Figure 5E and F*). Further in line with disruptions to downstream mitochondrial metabolism, we observe reduction of several acylcarnitine species at this age as well (*Figure 5—figure supplement 1A*). Although we observe decreased levels of the ATP-recycling metabolites phosphocreatine and creatine at P4, retinal ATP levels at this age are slightly but not significantly reduced (p = 0.5) (*Figure 5E*).

To further investigate possible glycolytic disruptions in P4 knockout retinas, we assayed the expression of a collection of glycolysis and TCA cycle enzymes in P4 knockout and control retinas using our RNA-sequencing dataset. Indeed, this analysis reveals broad transcriptional changes in 16 glycolytic and 4 TCA cycle enzymes in P4 knockout retinas (*Figure 5G*). Notably, two of these changes—upregulation of the aldehyde dehydrogenase *Aldh8a1* and downregulation of the TCA cycle enzyme *Ogdhl*—are also present in RNA-sequencing results from the pre-degenerative (E18.5) timepoint (*Figure 5—figure supplement 1B*). Finally, consistent with

disruption of glycolytic flux and reduced NADP levels in knockout retinas, mass spectrometry suggests possible disruption of the pentose phosphate pathway (PPP) evidenced by decreased levels of erythritol at this age (*Figure 5H*).

## NMNAT1 loss disrupts retinal purine metabolism and a subset of amino acids

As MSEA also indicated possible disruptions of purine metabolism and several amino acid pathways in P4 NMNAT1 knockout retinas (*Figure 5B*), we closely examined levels of a collection of metabolites representing most major nucleotide and amino acid metabolites in P4 knockout and control retinas (*Figure 5—figure supplement 2*). Interestingly, while levels of the pyrimidine nucleotide derivatives uracil, UDP, and cytidine were unchanged at this age, levels of the purine nucleotide precursor xanthine, guanine, and GMP were increased by ~150%, ~147%, and ~58%, respectively (*Figure 5—figure supplement 2A*). The sole affected pyrimidine nucleotide was cytosine, which showed an impressive ~352% increase in P4 knockout retinas as compared to controls (*Figure 5—figure supplement 2A*). In addition to defects in purine metabolism, we observe significant changes in a subset of 10 amino acids and amino acid derivatives in P4 knockout retinas including acetyl-asparagine and acetyl-lysine (*Figure 5—figure supplement 2B*).

Overall, these metabolomics results suggest specific disruptions to central carbon, purine nucleotide, and amino acid metabolism as potential causes for severe retinal degeneration in the absence of NMNAT1.

## Discussion
### NMNAT1-deficiency is associated with early and severe retinal degeneration involving multiple cell types

In this study, we demonstrate that retinal NMNAT1 deficiency in mice leads to severe degeneration of photoreceptor, bipolar, horizontal, and amacrine neurons soon after birth. In general, this phenotype is consistent with several recent studies reporting partial or complete ablation of retinal NMNAT1, which report gross retinal degeneration beginning within the first postnatal week and largely complete by one month of age (*Wang et al., 2017*; *Eblimit et al., 2018*). While the present study describes a relatively rapid timeline of NMNAT1-associated retinal degeneration, it furthers the characterization of this phenotype in two important ways: first, by assessing the survival of specific retinal neuron subtypes, and second, by systematically examining retinal cell death pathways triggered by NMNAT1 deletion.

Examination of specific cell types in NMNAT1 knockout retinas reveals that, while photoreceptors are likely the primary targets of NMNAT1-associated pathology, retinal bipolar, horizontal, and amacrine cells are also significantly affected by loss of NMNAT1 during retinal development, while ganglion cells appear unaffected until later stages. In particular, our results indicate an important distinction between the sensitivity of photoreceptor and non-photoreceptor neurons to NMNAT1 loss: while photoreceptors demonstrate an early and sustained transcriptional downregulation of cell-type-specific genes, near complete abrogation of recoverin and opsin expression, and rapid degeneration, non-photoreceptor cell types show largely unchanged transcriptional signatures, persistent cell-type marker expression, and a more gradual degeneration in NMNAT1 knockout retinas. These results stand partially in contrast to a recent study reporting ex vivo knockdown of *Nmnat1* in retinal explant cultures, which reported thinner INLs but no changes in numbers of HuC/D-expressing amacrine cells or PKCa-expressing bipolar cells in NMNAT1-deficient explants (*Kuribayashi et al., 2018*). However, this study did report reduced numbers of PNR-positive photoreceptor cells in NMNAT1-deficient explants, and differences may be explained by the fact that *Nmnat1* expression in explants was knocked down relatively late in development (E17.5)—by which time many non-photoreceptor cell types have already differentiated (*Cepko, 2014*). Alternatively, the relative persistence of non-photoreceptor cells in our model may indicate that degeneration of bipolar, horizontal, and amacrine neurons is a secondary effect of massive photoreceptor death, a possibility potentially supported by evidence of INL degeneration in a photoreceptor-specific NMNAT1 knockout mouse model (*Wang et al., 2017*).

The generation and characterization of retinal cell-type-specific NMNAT1 knockout models is warranted to examine whether this differential sensitivity to NMNAT1 loss is cell-autonomous.

While we were unable to successfully determine retinal NMNAT1 distribution using our polyclonal NMNAT1 antibody, previously published results showing RT-qPCR of *Nmnat* levels in flow-sorted rod photoreceptors suggest that NMNAT1 is the predominantly expressed NMNAT enzyme in rod photoreceptors (*Kuribayashi et al., 2018*). Combined with our data confirming a lack of transcriptional upregulation of *Nmnat2* or *Nmnat3* in NMNAT1 knockout retinas (*Figure 3—figure supplement 1*), one possible explanation for the cell-type-specific degeneration we observe is that relatively higher levels of NMNAT2/3 in inner retinal neurons can partially compensate for the $NAD^+$ deficit caused by loss of NMNAT1. Recent results in a mutant NMNAT1 mouse model indicating relative depletion of $NAD^+$ in the retina but not other tissues (*Greenwald et al., 2021*) are perhaps consistent with such a hypothesis. Overall, our results demonstrate that NMNAT1 deficiency during retinal development affects multiple cell types beyond photoreceptors and suggest retinal cell-type-specific requirements of $NAD^+$ metabolism to be further investigated.

## NMNAT1-deficient retinas activate multiple cell death pathways

Previous reports of cell death in NMNAT1-deficient retinas center on two potentially contrasting mechanisms: the aforementioned ex vivo study outlined a role for *Noxa* and *Fas*-associated, caspase-3 dependent apoptosis in NMNAT1-deficient explants (*Kuribayashi et al., 2018*), while a recent study found the death of mature photoreceptors after global NMNAT1 deletion to be solely dependent on the presumed non-apoptotic NADase SARM1 (*Sasaki et al., 2020b*). Using histological and comprehensive transcriptomic approaches, we demonstrate involvement of caspase-3 associated apoptosis in NMNAT1 knockout retinas characterized by an early and sustained upregulation of *Noxa* and deregulation of several other apoptosis-pathway genes. We extend these results by showing histological evidence of a distinct, caspase-3-independent cell death pathway which constitutes a significant portion of observed cell death and closely follows caspase-3-dependent apoptosis throughout the timepoints tested in our model.

Interestingly, using a recently validated metabolic marker of SARM1 activity (cADPR) (*Sasaki et al., 2020a*), we do not detect SARM1 involvement at pre- or post-degenerative timepoints in our model. That we do not find evidence of SARM1 involvement, even in the presence of increased levels of its potent activator NMN (*Zhao et al., 2019*; *Figley et al., 2021*) is unexpected but not implausible, especially considering the fact that the report implicating SARM1 in NMNAT1-associated degeneration deleted NMNAT1 in mature mice (*Sasaki et al., 2020b*). Indeed, many apoptosis effectors (*Casp3*, *Casp9*, *Apaf1*, *Bcl* family members) active in the developing mammalian retina are subsequently downregulated in mature retinas, necessitating alternative death pathways for handling pathological insults at these ages (*Donovan and Cotter, 2002*; *Doonan et al., 2003*; *Donovan et al., 2006*). Considering these results, it is wholly possible that in mature retinas with insufficient expression of necessary apoptotic and/or pyroptotic machinery, NMNAT1-associated retinal degeneration proceeds through SARM1—in such a case, it is important to note that model systems with embryonic or germline deletion or mutation of NMNAT1 are typically more representative of patients with disease-linked mutations in NMNAT1. On the other hand, it cannot be conclusively ruled out that excess cADPR produced by an active SARM1 in our model is rapidly metabolized, and recently described links between SARM1 and both pyroptosis and apoptosis (*Mukherjee et al., 2015*; *Carty et al., 2019*) leave open the possibility of cooperation between these cell death pathways in NMNAT1 knockout retinas.

Our transcriptomics results indicate dysregulation of several pyroptosis and necroptosis-related genes, notably including the entire 'canonical inflammasome' (*Casp1*, *Nlrp3*, and *Pycard*), as well as the toll-like receptors *Tlr2* and *Tlr4*, at both timepoints (*Figure 4*). The distinct presence of nuclear pyknosis in AC3⁻ dying cells—which is generally incompatible with necroptosis but documented in pyroptotic cells (*Vandenabeele et al., 2010*; *Murakami et al., 2012*; *Miao et al., 2011*)—lends support to pyroptosis as a significant driver of cell death in NMNAT1-deficient retinas. Intriguingly, we do not detect proteolytic cleavage of gasdermin D—a common marker of pyroptosis—in P4 NMNAT1 knockout retinas (*Figure 4—figure supplement 1*). However, recent results indicate that NLRP3 is, under certain circumstances, capable of being activated independently of gasdermin D (*Gutierrez*

*et al., 2017*). Considering a recent report suggesting involvement of the PARP1-associated 'parthanatos' cell death pathway in NMNAT1 mutant retinas (*Greenwald et al., 2021*), we did not detect accumulation of poly ADP-ribose (PAR) in AC3⁻ pyknotic nuclei (data not shown), arguing against involvement of this pathway under these conditions.

In sum, we show that retinal NMNAT1 loss activates multiple cell death pathways, which contextualizes the degeneration of multiple cell types in our model and may explain the severity of NMNAT1-associated retinal degeneration in this study and others. Although the model presented here differs in important ways from NMNAT1-mutant LCA animal models, it does recapitulate some aspects of NMNAT1-linked LCA including particularly severe central retinal defects (*Kumaran et al., 2017*). As recent studies have discovered noncoding mutations, copy number variations, and exon duplications in NMNAT1 causing severe reduction of NMNAT1 expression in patients with ocular and extra-ocular pathologies (*Coppieters et al., 2015*; *Bedoni et al., 2020*), understanding the mechanisms of retinal degeneration in an NMNAT1 knockout model is of potential clinical significance.

## Retinal NMNAT1 loss causes diverse metabolic disruptions

Several recent studies examine levels of select metabolites in mature NMNAT1-deficient retinas (*Sasaki et al., 2020b*), or more broadly examine tissue metabolomes after perturbation of the NMN-synthesizing enzyme NAMPT (*Lin et al., 2016*; *Oakey et al., 2018*; *Lundt et al., 2021*), but to date there have been no comprehensive metabolomic studies on NMNAT1-deficient tissues. Our metabolomics results approximate that NMNAT1 synthesizes ~40% of the total retinal NAD⁺ pool, which is generally consistent with a previous model (*Sasaki et al., 2020b*). In addition, our results strongly suggest that—via specific disruptions to central carbon, nucleotide, and amino acid metabolism—depletion of NAD⁺ synthesized in the nucleus disrupts multiple non-nuclear metabolic pathways in the retina. Whether these observations reflect a direct export of nuclear NAD⁺ to cytosolic and mitochondrial retinal compartments versus an indirect effect on cytosolic and mitochondrial cellular processes (by way of NAD⁺-dependent gene regulation, for instance) is a topic for further study.

Impaired glycolytic flux appears to be a more general feature of tissue NAD⁺ depletion, as studies reporting NAMPT inhibition or deletion in projection neurons and skeletal muscle myotubes report accumulation of glycolytic metabolites upstream of GAPDH (*Oakey et al., 2018*; *Lundt et al., 2021*). Interestingly, Lundt et al. present evidence of reversed glycolytic flux in NAMPT-inhibited myotubes, an effect which we believe may explain decreased levels of G3P and DHAP in our model. Notably, unbiased LC-MS/MS and GC-MS analyses of rod-photoreceptor-specific NAMPT knockout retinas showed signatures of mitochondrial metabolic defects—which we observe in our model as well—but detected limited evidence of glycolytic impairment (*Lin et al., 2016*). This suggests that retinal NMNAT1 and NAMPT depletion, despite both lowering total retinal NAD⁺ levels, produce distinct metabolic phenotypes.

Retinal neurons—and retinal photoreceptors in particular—are relatively unique in their dependence on aerobic glycolysis (the 'Warburg Effect') during both proliferative and differentiated states (*Agathocleous et al., 2012*; *Ng et al., 2015*; *Chinchore et al., 2017*). Our finding that retinal NMNAT1 loss is detrimental to glycolysis thus offers a potential explanation for the cell-type-specific degeneration which we observe: in particular, previous results indicating that differentiation induces an increased reliance on mitochondrial OXPHOS relative to glycolysis in the retina (*Agathocleous et al., 2012*) might explain why ganglion and amacrine cells—which differentiate relatively early—are less sensitive to NMNAT1 loss in our model than the later born bipolar and photoreceptor cells. Indeed, even in mature retinas, glycolytic perturbations specifically affect photoreceptor health and survival (*Chinchore et al., 2017*; *Zhang et al., 2020*; *Sinha et al., 2021*). Furthermore, recent results linking glycolytic impairment to NLRP3 activation (*Sanman et al., 2016*) provide a potential explanation of non-apoptotic cell death which we observe in NMNAT1 knockout retinas.

In addition to glycolytic impairment, we detect specific defects in purine nucleotide and amino acid metabolic pathways in NMNAT1 knockout retinas. As a particularly proliferative tissue, the retina is thought to be highly reliant on adequate nucleotide and amino acid pools to support

transcription and translation of cell-specific machinery (*Etingof, 2001*; *Ng et al., 2015*). Some of the metabolic changes which we observe—for instance, accumulation of the purine precursor xanthine and the amino acid aspartate—appear to be more widely associated with NAD$^+$ insufficiency or retinal degeneration (*Du et al., 2014*; *Lin et al., 2016*; *Oakey et al., 2018*). On the other hand, we also identify a collection of metabolic changes in these pathways which are not reported in NAMPT-deficient retinas (*Lin et al., 2016*), potentially explaining differences in retinal phenotypes between these two models.

## A potential role for NMNAT1 in photoreceptor terminal differentiation

The tightly coordinated and stereotypical differentiation of retinal neuron subtypes from a common progenitor pool has been extensively studied—complementing classical birth-dating studies, recent investigations have begun to explore the massive epigenetic regulation necessary for the development of the mammalian retina (*Swaroop et al., 2010*; *Aldiri et al., 2017*; *Raeisossadati et al., 2021*). One of the most surprising findings of the present study is an early and sustained transcriptional downregulation of a subset of photoreceptor- and synapse-specific genes in NMNAT1 knockout retinas. Beyond these transcriptional disruptions, we show near complete absence of several crucial rod and cone-specific proteins in P4 knockout retinas, as well as early and sustained mis-expression of OTX2—a crucial differentiation factor—in the absence of NMNAT1. While NMNAT1 has previously been implicated in gene regulation through direct interaction with SIRT1 and PARP1 at gene promoters (*Zhang et al., 2012*; *Song et al., 2013*) and retinal NMNAT1 knockdown was shown to influence apoptotic gene expression by potentially modulating histone acetylation (*Kuribayashi et al., 2018*), no in vivo role for NMNAT1 in retinal developmental gene regulation has yet been described.

Photoreceptors are among the last retinal cell types to fully develop and are generated in two broad phases: an early cell-fate commitment mediated by several well-characterized transcription factors including NOTCH1, PAX6, and SIX3, and a later phase ('terminal differentiation') comprising expression of a host of specialized phototransduction genes and growth of light-sensing cellular structures, mediated in part by the transcription factors OTX2, NRL, and CRX (*Swaroop et al., 2010*; *Brzezinski and Reh, 2015*; *Daum et al., 2017*). Interestingly, we do not observe transcriptional changes in NOTCH1, PAX6, SIX3, or related genes in NMNAT1 knockout retinas at E18.5. This fact, combined with grossly normal retinal morphology and comparable numbers of PHH3-positive proliferating cells in P0 knockouts, suggests that NMNAT1 is required for terminal differentiation but not early specification or proliferation of retinal photoreceptor cells. Increased abundance of acetyl-lysine in knockout retinas on mass spectrometry (*Figure 5—figure supplement 2B*) and GO enrichment of several genes associated with DNA methylation, epigenetic regulation, and chromatin silencing in P4 knockout retinas (*Figure 3—figure supplement 2*) support a potential role for NMNAT1 in the epigenetic regulation of photoreceptor terminal differentiation, a phenomenon recently shown to feature genome-wide methylation, demethylation, and acetylation events (*Aldiri et al., 2017*; *Seritrakul and Gross, 2017*, *Rhee et al., 2012*). Importantly, although cell markers for bipolar, horizontal, and amacrine retinal neurons suggest proper specification and rapid degeneration of these cells in NMNAT1 knockouts (*Figure 2*), more extensive studies focusing on early specification markers are needed before a role for NMNAT1 in the differentiation of these cell types can be ruled out.

In conclusion, this study presents a comprehensive evaluation of NMNAT1-associated retinal dysfunction and suggests crucial roles for nuclear NAD$^+$ in the proper development of the mammalian retina. Extending previous results, we demonstrate that retinal NMNAT1 loss during development affects retinal cell types beyond photoreceptors, and we propose a yet-undescribed role for NMNAT1 in photoreceptor terminal differentiation and subsequent survival. While we provide evidence that the early and severe retinal degeneration associated with NMNAT1 loss involves multiple cell types and death pathways, it appears that this severe phenotype stems from two major problems: (1.) metabolic defects likely caused by insufficient NAD$^+$ for retinal proliferative metabolism, and (2.) gene regulation defects potentially caused by insufficient nuclear NAD$^+$ in developing photoreceptors. Considering links between metabolic state and differentiation in the retina and recently discovered roles of compartmentalized NAD$^+$ in non-retinal cell differentiation (*Agathocleous et al., 2012*; *Agathocleous and Harris, 2013*; *Ryu et al., 2018*), these two problems may not be mutually exclusive. Further study

of NMNAT1-associated retinal dysfunction should focus on evaluating the relative contributions of metabolic and genetic deficits to the overall pathology and testing the hypothesis that NMNAT1 functions to integrate energy metabolism and gene regulation, both in the retina and in other cellular contexts.

## Materials and methods

**Key resources table**

| Reagent type (species) or resource | Designation | Source or reference | Identifiers | Additional information |
|---|---|---|---|---|
| Gene (*Mus musculus*) | *Nmnat1* | | MGI:1913704 | |
| Genetic reagent (mouse, male and female) | *Nmnat1*<sup>fl/fl</sup> | **Conforti et al., 2011** | | Nmnat1 gene exons 1 and 2 floxed |
| Genetic reagent (mouse, male and female) | Six3-Cre | **Christiansen et al., 2011** | | Expresses Cre in developing eye bud (~E12.5) |
| Cell line (*Homo- sapiens*) | HEK293T | ATCC | Cat# CRL-3216 | |
| Transfected construct (*M. musculus*) | FLAG-NMNAT1 | Genscript | Cat# OMu17664D | FLAG-tagged mouse NMNAT1 construct |
| Transfected construct (*M. musculus*) | FLAG-NMNAT2 | Genscript | Cat# OMu16562D | FLAG-tagged mouse NMNAT2 construct |
| Antibody | Anti-NMNAT1 (rabbit polyclonal) | this paper | | WB (1:1000) See Materials and methods |
| Antibody | Anti-beta tubulin (mouse monoclonal) | Sigma | Cat# T5201, RRID: AB_609915 | WB (1:3000) |
| Antibody | Anti-FLAG (mouse monoclonal) | Sigma | Cat# F1804, RRID: AB_262044 | WB (1:2000) |
| Antibody | Anti-BRN3A (mouse monoclonal) | Santa Cruz | Cat#: sc-8429, RRID: AB_626765 | IF (1:150) |
| Antibody | Anti-calretinin (rabbit polyclonal) | Millipore Sigma | Cat# AB5054 RRID: AB_2068506 (discontinued) | IF (1:500) |
| Antibody | Anti-CHX10 (mouse monoclonal) | Santa Cruz | Cat# sc-365519, RRID: AB_10842442 | IF (1:50) |
| Antibody | Anti-recoverin (rabbit polyclonal) | Sigma | Cat# AB5585, RRID: AB_2253622 | IF (1:500) |
| Antibody | Anti-rhodopsin (4D2) (mouse monoclonal) | R.Molday, Univ. British Columbia | | IF (1:500) |
| Antibody | Anti-synaptophysin (rabbit monoclonal) | Thermo Fisher | Cat# MA5-14532, RRID: AB_10983675 | IF (1:1000) |
| Antibody | Anti-active caspase 3 (rabbit polyclonal) | R&D Systems | Cat# AF835, RRID: AB_2243952 | IF (1:500) |
| Antibody | Anti-OTX2 (rabbit polyclonal) | ProteinTech | Cat# 13497–1-AP, RRID: AB_2157176 | IF (1:500) |
| Antibody | Anti-phospho histone H3 (rabbit polyclonal) | Cell Signaling Technologies | Cat# 9701, RRID: AB_331535 | IF (1:1000) |
| Antibody | Anti-phosducin (sheep polyclonal) | **Sokolov et al., 2004** | | IF (1:1000) |
| Antibody | Anti-M opsin (rabbit polyclonal) | Millipore Sigma | Cat# AB5045, RRID: AB_177456 | IF (1:1000) |
| Antibody | Anti-calbindin (mouse monoclonal) | Swant | Cat# 300, RRID: AB_10000347 | IF (1:500) |
| Antibody | Anti-rabbit Alexa Fluor- 568 (goat polyclonal) | Invitrogen | Cat# A-11011, RRID: AB_143157 | IF (1:1000) |

*Continued on next page*

*Continued*

| Reagent type (species) or resource | Designation | Source or reference | Identifiers | Additional information |
|---|---|---|---|---|
| Antibody | Anti-mouse Alexa Fluor- 488 (goat polyclonal) | Invitrogen | Cat# A-11001, RRID: AB_2534069 | IF (1:1000) |
| Antibody | Anti-rabbit Alexa Fluor 680 (goat polyclonal) | Invitrogen | Cat# A-21076, RRID: AB_2535736 | WB (1:50,000) |
| Antibody | Anti-mouse DyLight 800 | Invitrogen | Cat# SA5-10176, RRID: AB_2556756 | WB (1:50,000) |
| Antibody | Fluorescein Avidin D | Vector Laboratories | Cat# A-2001–5 | IF (1:500) For visualizing biotinylated PNA stain (below) |
| Other | Biotinylated peanut agglutinin (PNA) | Vector Laboratories | Cat# B-1075–5 | IF (1:500) |
| Other | 4',6-diamidino-2-phenylindole (DAPI) | Thermo Fisher | Cat# D1306, RRID: AB_2629482 | IF (1:2000) |
| Software, algorithm | Fiji | *Schindelin et al., 2012* | RRID:SCR_002285 | |
| Software, algorithm | FastQC | | RRID:SCR_014583 | https://www.bioinformatics.babraham.ac.uk/projects/fastqc/ |
| Software, algorithm | Bbduk | | RRID:SCR_016969 | http://sourceforge.net/projects/bbmap/ |
| Software, algorithm | HISAT2 2.1.0 | *Kim et al., 2015* | RRID:SCR_015530 | |
| Software, algorithm | StringTie 1.3.6 | *Pertea et al., 2015*; *Pertea et al., 2016* | RRID:SCR_016323 | |
| Software, algorithm | DeSeq2 R Package | *Love et al., 2014* | RRID:SCR_015687 | |

## Animal model generation, husbandry, and genotyping

*Nmnat1*fl/fl mice described previously (*Conforti et al., 2011*) were thoroughly backcrossed with wild-type 129/SV-E mice (Charles River Laboratories, Wilmington, MA) prior to analyses. Conditional knockout mice were generated by crossing *Nmnat1*fl/fl mice with transgenic mice expressing Cre recombinase under a *Six3* promoter (*Six3-Cre*) (*Christiansen et al., 2011*). Crosses yielded hetero-zygous *Nmnat*fl/wt and *Nmnat*fl/wt;*Six3-Cre* offspring, which were further crossed with *Nmnat1*fl/fl mice to yield conditional knockout (*Nmnat*fl/fl;*Six3-Cre*) and littermate control (*Nmnat1*fl/fl) mice at approx-imately Mendelian ratios. Experimental animals were periodically backcrossed with wild-type 129/SV-E mice to maintain genetic integrity. Animals were maintained under standard 12 hr light/dark cycles with food and water provided ad libitum. All experimental procedures involving animals were approved by the Institutional Animal Care and Use Committee (IACUC) of West Virginia University.

Animals were genotyped using polymerase chain reaction (PCR) of genomic DNA from ear punch biopsies. Primer sequences for detection of *Six3-Cre* transgene and *Nmnat1* 5' and 3' loxP sites are listed in the **Key Resources Table**, and primers were added to the PCR reaction at a final concentra-tion of 0.75 µM. The thermocycling conditions were 95 °C for 2 min, 35 cycles of 95 °C for 30 s, 58 °C for 30 s, 72 °C for 45 s, and a final extension step of 72 °C for 5 min.

## NMNAT1 antibody generation

A rabbit polyclonal antibody against amino acids 111–133 of mouse NMNAT1 (sequence CSYPQSSP ALEKPGRKRKWADQK) was generated by Pacific Immunology Corp. (Pacific Immunology, Ramona, CA). The affinity-purified antibody was confirmed to recognize NMNAT1 in cell culture and retinal lysate (*Figure 1—figure supplement 1*), and did not recognize overexpressed FLAG-NMNAT2 (data not shown).

## Mammalian cell culture and transfection

HEK293T cells were maintained in 1:1 DMEM/F-12 culture media (Thermo Fisher Scientific, Waltham, MA) in a sterile incubator at 37 °C and 5% $CO_2$. Transient transfection of FLAG-NMNAT1 and

FLAG-NMNAT2 constructs was performed at ~60% confluence using TransIT-LT1 transfection reagent (Mirus Bio, Madison, WI) according to manufacturer's instructions. Cells were harvested 48 hr after transfection on ice with 1X Hank's balanced salt solution (HBSS) (Thermo Fisher Scientific) and cell pellets were processed for western blotting as described below. As these cells were used only briefly for antibody validation purposes, we were unable to authenticate the identity of these cells.

## Retinal histology and thickness quantification

To evaluate gross retinal histology, mice at the indicated ages were sacrificed and their eyes were gently removed into 1 ml Excalibur's Alcoholic z-fix (Excalibur Pathology Inc, Norman, OK). Subsequent fixation, paraffin embedding, sectioning, and hematoxylin and eosin (H&E) staining were performed by Excalibur Pathology. H&E-stained sections were imaged on a Nikon Eclipse Ti microscope with DS-Ri2 camera (Nikon Instruments, Melville, NY). Retinal thickness was measured using Nikon NIS-Elements software at four equidistant points along the outer retinal edge to either side of the optic nerve, where retinal thickness is defined as the length of a line orthogonal to the outer retinal edge and terminating at the inner retinal edge. Thickness was quantified using six technical replicates per animal and three biological replicates per genotype.

## Western blotting

For determination of protein levels by western blot, retinas were homogenized in cold lysis buffer (1 X phosphate-buffered saline pH 7.4 (Thermo Fisher Scientific), Pierce protease inhibitor, EDTA-free (Thermo), Calbiochem phosphatase inhibitor cocktail (EMD Biosciences, La Jolla, CA)) using a Microson ultrasonic cell disruptor. Protein concentration was determined using a Nanodrop ND-1000 spectrophotometer (Thermo Fisher Scientific), after which Laemmli sample buffer was added to a final concentration of 1X (2% SDS, 0.05 M Tris-HCl pH 6.8, 10% glycerol, 1%beta-mercaptoethanol, 2 mM dithiothreitol (DTT), 0.001% bromphenol blue) and samples were boiled for 5 min. Equal amounts of protein were separated on anykD mini-PROTEAN TGX polyacrylamide gels (Bio-Rad, Hercules, CA) and transferred to an Immobilon-FL PVDF membrane (Millipore, Burlington, MA). Membranes were blocked with Odyssey blocking buffer (LI-COR Biosciences, Lincoln, NE) for 1 hr at room temperature and incubated with primary antibodies diluted in 50/50 blocking buffer/1 X PBST (1 X PBS + 0.1% Tween-20) at 4 °C overnight on a bidirectional rotator. A list of primary antibodies, sources, and dilutions can be found in the **Key Resources Table**. Following primary antibody incubations, membranes were washed in 1 X PBST and incubated with goat anti-rabbit Alexa Fluor 680 and/or goat anti-mouse DyLight 800 secondary antibodies (Thermo Fisher Scientific) for 1 hr at room temperature. Membranes were subsequently washed in 1 X PBST and imaged using an Odyssey Infrared Imaging System (LI-COR Biosciences).

## Quantitative reverse transcriptase-PCR (RT-qPCR)

To determine relative gene expression using RT-qPCR, retinas were collected and total RNA was isolated as described below. cDNA was synthesized with a RevertAid cDNA Synthesis Kit (Thermo Fisher Scientific) per manufacturer instructions, starting with 1 µg total RNA/sample and using random hexamer primers. Primers for targets of interest were designed using NCBI Primer-BLAST (https://www.ncbi.nlm.nih.gov/tools/primer-blast/) to straddle at least one intron of length >500 bp. qPCR reactions were performed using Brilliant II SYBR Green qPCR master mix with low ROX (Agilent Technologies, Cedar Creek, TX) and monitored on a Stratagene Mx3000P cycler (Agilent Technologies). Prior to use, primers were validated (by examining melt curves and agarose gel electrophoresis) to amplify single products of expected size. Three technical replicates per target per animal were performed, and averages from three animals per genotype are reported. All expression values were normalized to the geometric average of three housekeeping genes (*Hmbs*, *Ppia*, and *Ywhaz*). Primer sequences are included in *Supplementary file 1*.

## Immunofluorescent staining

After euthanasia, eyes were gently removed, punctured, and immersed in 4% paraformaldehyde fixative (4% paraformaldehyde in 1 X PBS) (Electron Microscopy Sciences, Hatfield, PA) for 15 min, after which the cornea was removed and the eye was fixed for an additional 45 min at room temperature with gentle agitation. Subsequently, eyes were washed in 1 X PBS and incubated in a dehydration

solution (20% sucrose in 1 X PBS) for at least 12 hr at 4 °C. After dehydration, samples were incubated in a 1:1 mixture of 20% sucrose and Tissue-Tek O.C.T. compound (Sakura Finetek, Torrance, CA) for at least 1 hr before being transferred to 100% O.C.T. and flash frozen. 16 µm sections were cut using a Leica CM1850 cryostat (Leica Biosystems, Nussloch, Germany) and mounted on Superfrost Plus slides (Fisher Scientific). For immunofluorescent staining, retinal sections were briefly rinsed with 1 X PBS and incubated in blocking buffer (10% normal goat serum, 0.5% Triton X-100, 0.05% sodium azide in 1 X PBS) for 1 hr at room temperature. Following blocking, sections were incubated with the indicated primary antibodies (diluted in buffer containing 5% normal goat serum, 0.5% Triton X-100, 0.05% sodium azide in 1 X PBS) overnight at 4 °C. The next day, sections were washed with 1 X PBST and incubated with DAPI nuclear stain (Thermo Fisher Scientific), goat anti-rabbit Alexa Fluor-568 and/or goat anti-mouse Alexa Fluor-488 (Thermo Fisher Scientific) secondary antibodies for 1 hr at room temperature. For antibody information and dilutions, see the **Key Resources Table**. Finally, sections were washed, cover-slipped with Prolong Gold antifade reagent (Thermo Fisher Scientific), and imaged on a Nikon Eclipse Ti laser scanning confocal microscope with C2 camera (Nikon Instruments). Experimental and control sections were imaged using identical laser intensity and exposure settings. All fluorescent images represent maximum intensity z-projections generated using ImageJ with the Bio-Formats plugin (https://imagej.net/Bio-Formats).

## Retinal cell type and caspase-positive cell quantification

To estimate numbers of pyknotic and caspase-3 positive/negative cells in retinas of knockout and control mice, pyknotic nuclei were manually counted in $212.27 \times 212.27\ \mu m^2$ regions of maximum intensity projection fluorescent images from the central retina. Once these pyknotic nuclei were identified, the number also labeled with active-Caspase-3 were manually counted. Counts from two technical replicates per animal were averaged, and averages from three animals per genotype are reported. Counts of retinal cell subtypes were estimated by manually counting marker-positive cells in $318.2 \times 318.2\ \mu m^2$ regions of central retina, averaged between two technical replicates per animal and three animals per genotype. All counts were obtained using ImageJ. For antibody information, refer to the **Key Resources Table**.

## Retinal RNA extraction, sequencing, and analysis

Following euthanasia and eye enucleation, retinae were quickly dissected and transferred into ultra-sterile microcentrifuge tubes containing a small amount of Trizol reagent (Thermo Fisher Scientific), and flash frozen on dry ice. Total RNA was extracted by homogenizing thawed samples with a hand-held homogenizer and incubating for 5 min at room temperature. 20 µl chloroform was added to each sample, samples were briefly vortexed, incubated at room temperature for 4 min, and spun at 12,000 rpm for 10 min at 4 °C. The aqueous layers were removed to separate tubes containing 50 µl isopropanol and incubated at room temperature for 10 min with occasional agitation. Finally, samples were again spun at 12,000 rpm for 10 min at 4 °C, supernatants were removed, and pellets were washed three times with 75% ethanol, dried, and resuspended in DEPC-treated water. Whole-transcriptome sequencing was performed by Macrogen Corp. (Macrogen USA, Rockville, MD) using an Illumina TruSeq Stranded mRNA Library Prep Kit on a NovaSeq6000 S4 to a depth of 100 M total reads per sample. Read quality was verified using FastQC (https://www.bioinformatics.babraham.ac.uk/projects/fastqc/) and adapters were trimmed using the Bbduk utility of the BBTools package (http://sourceforge.net/projects/bbmap/). Read alignment was performed using HISAT2 2.1.0 (*Kim et al., 2015*) and transcripts were assembled and quantified using StringTie 1.3.6 (*Pertea et al., 2015*; *Pertea et al., 2016*). Differentially expressed genes were identified using the DESeq2 R package (*Love et al., 2014*).

## Targeted steady-state metabolomics using LC-MS/MS

Following euthanasia and eye enucleation, retinae were quickly dissected into Hank's Balanced Salt Solution (HBSS) and flash frozen on liquid nitrogen. Metabolites were extracted according to previously described protocols (*Grenell et al., 2019*; *Yam et al., 2019*). Metabolite extracts were analyzed by a Shimadzu LC Nexera X2 UHPLC coupled with a QTRAP 5500 LC-MS/MS (AB Sciex). An ACQUITY UPLC BEH Amide analytic column (2.1 × 50 mm, 1.7 µm) (Waters Corp., Milford, MA) was used for chromatographic separation. The mobile phase was (A) water with 10 mM ammonium acetate (pH 8.9)

and (B) acetonitrile/water (95/5) with 10 mM ammonium acetate (pH 8.2) (All solvents were LC–MS Optima grade from Fisher Scientific). The total run time was 11 min with a flow rate of 0.5 ml/min and an injection volume of 5 µl. The gradient elution was 95–61% B in 6 min, 61–44% B at 8 min, 61–27% B at 8.2 min, and 27–95% B at 9 min. The column was equilibrated with 95% B at the end of each run. The source and collision gas was $N_2$. The ion source conditions in positive and negative mode were: curtain gas (CUR) = 25 psi, collision gas (CAD) = high, ion spray voltage (IS) = 3800/–3800 volts, temperature (TEM) = 500 °C, ion source gas 1 (GS1) = 50 psi, and ion source gas 2 (GS2) = 40 psi. Each metabolite was tuned with standards for optimal transitions. D4-nicotinamide (Cambridge Isotope Laboratories, Tewksbury, MA) was used as an internal standard. The extracted MRM peaks were integrated using MultiQuant 3.0.3 software (AB Sciex, Concord, ON, CA). Additional metabolite parameters are available in *Supplementary file 1*.

## Statistical analyses

Sample number (n) is defined as number of animals per genotype. Specific statistical tests and sample sizes are indicated in figure legends and/or in the Transparent Reporting Form. Where applicable, p-value adjustments for multiple comparisons were performed and indicated, and reported as 'q' values. Across all figures, statistical significance is defined as $p < 0.05$ (or $q < 0.05$, where applicable). Experimenters were not blinded to treatments.

## Acknowledgements

We thank Dr. Laura Conforti for the *Nmnat1*flox/flox animals, and current and past members of the Kolandaivelu lab for constant support. This work was supported by bridge funding to SK and the National Institutes of Health (RO1EY028959 to SK).

## Additional information

### Funding

| Funder | Grant reference number | Author |
| --- | --- | --- |
| West Virginia University | Bridge Funding | Saravanan Kolandaivelu |
| National Institutes of Health | RO1EY028959 | Saravanan Kolandaivelu |

The funders had no role in study design, data collection and interpretation, or the decision to submit the work for publication.

### Author contributions

David Sokolov, Conceptualization, Data curation, Formal analysis, Investigation, Methodology, Software, Validation, Visualization, Writing - original draft, Writing – review and editing; Emily R Sechrest, Investigation, Validation, Writing – review and editing; Yekai Wang, Data curation, Formal analysis, Investigation, Software; Connor Nevin, Investigation; Jianhai Du, Methodology, Resources, Supervision; Saravanan Kolandaivelu, Conceptualization, Funding acquisition, Investigation, Methodology, Project administration, Resources, Supervision, Validation, Writing – review and editing

### Author ORCIDs

David Sokolov http://orcid.org/0000-0001-6774-4133
Saravanan Kolandaivelu http://orcid.org/0000-0002-8552-7850

### Ethics

This study was performed in strict accordance with the recommendations in the Guide for the Care and Use of Laboratory Animals of the National Institutes of Health. All of the animals were handled according to approved institutional animal care and use committee (IACUC) protocols of West Virginia University. The protocol was approved by the Institutional Animal Care and Use Committee of West Virginia University (Protocol #1603001820).

Decision letter and Author response
Decision letter https://doi.org/10.7554/eLife.71185.sa1
Author response https://doi.org/10.7554/eLife.71185.sa2

## Additional files

### Supplementary files
• Supplementary file 1. Primer sequences for genotyping and RT-qPCR experiments and Mass spectrometry standards and parameters.
• Transparent reporting form

### Data availability
Sequencing data have been deposited in GEO under accession code GSE178312. All other data generated or analysed during this study are included in the manuscript and supporting files. Source data files have been provided for all figures.

The following dataset was generated:

| Author(s) | Year | Dataset title | Dataset URL | Database and Identifier |
|---|---|---|---|---|
| Sokolov D, Kolandaivelu S | 2021 | Nuclear NAD+-biosynthetic enzyme NMNAT1 facilitates development and early survival of retinal neurons | http://www.ncbi. nlm.nih.gov/geo/ query/acc.cgi?acc= GSE178312 | NCBI Gene Expression Omnibus, GSE178312 |

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
