## [Editor Report]

Mutations in the gene encoding the NMNAT1 enzyme cause Leber congenital amaurosis type 9 (LCA9), a blinding disease. Using conditional inactivation of the mouse gene in the retina, this study extends previous observations on the requirement of this enzyme for photoreceptors homeostasis. The study not only shows NMNAT1 is involved in photoreceptor terminal differentiation but also provides evidence that the survival of other cell types depends on this enzyme. It also shows that NMNAT1 deficiency leads to the activation of different cell death pathways and causes metabolic defects in retinal cells. The study thus provides a better picture of the retinal defects that may underlie LCA9 in humans.

---

## [Decision Letter]

**Decision letter after peer review:**

Thank you for submitting your article "Nuclear NAD+-biosynthetic enzyme NMNAT1 facilitates survival of developing retinal neurons" for consideration by *eLife*. Your article has been reviewed by 3 peer reviewers, one of whom is a member of our Board of Reviewing Editors, and the evaluation has been overseen by Marianne Bronner as the Senior Editor. The reviewers have opted to remain anonymous.

Essential revisions:

1) The authors need to experimentally address the potential role of NMNAT1 in retinal progenitor differentiation, with especial consideration of amacrine, photoreceptor and bipolar progenitors. The reviewers share the opinion that the data presented in the manuscript could be the result of defective specification/differentiation and not only linked to degeneration as the authors claim. Thus, staining for progenitor markers and proliferation antigens at embryonic and postnatal stages is required. Any other approach aimed at addressing this issue will be equally valid.

2) Analysis of cone, amacrine and Müller cell specification/differentiation should also be included as morphological and RNA-seq data suggests possible decrease of their specification/differentiation.

3) A better description of the knockout efficiency with the Cre-lox system used and its comparison with those used in the other studies to investigate NMNAT1 requirement in the retina is needed.

4) Statistical analysis of the data should be revised in Figure 1 and added in Figure 5.

5) A number of different studies have used similar approaches (retina-specific NMNAT1 KO or KD, transcriptomics, and metabolomics) to shed light on NMNAT1 function in the retina. The authors need to clarify what are the knowledge gaps they are trying to fill in and should discuss better their findings in relation to the existing data (i.e. in relation to those of Eblimit et al., 2018 and Kuribayashi et al., 2018).

6) The authors may also consider rescuing the phenotype of the CKO retina by the administration of NAD or NMN, if experimentally feasible.

*Reviewer #1:*

Mutations in the gene nicotinamide mononucleotide adenylyltransferase-1 (NMNAT1) causes a specific type of severe and early onset retinal degeneration known as Leber congenital amaurosis type 9. By conditional inactivating the NMNAT1 gene in the mouse eye bud, this study extends on previous reports showing that NMNAT1 is required for postnatal survival of photoreceptors and specific retinal interneurons. Using transcriptomic and metabolomic approaches, the authors provide novel evidence that retinal degeneration is linked to the disruption of carbon, purine and amino acid metabolism and to the activation of distinct cell death pathways, including apoptosis, pyroptosis and necroptosis.

Although well conducted the study does not consider the possible role of NMNAT1 in retinal progenitor differentiation. This is an important point because the phenotypic characterization of the retina at postnatal development suggest that possibility that amacrine, photoreceptor and bipolar cells might be generated in lower number besides undergoing later degeneration. This point needs to be addressed. Furthermore, the authors fell short of discussing in the details their data in comparison with those of Eblimit et al., 2018, who have use different Cre line to delete NMNAT1 at slightly later stages of development or only in photoreceptors.

Figure 2. When comparing the number of amacrine and bipolar cells in the control groups, it is clear that both populations undergo an increase in size between P4 and P10. The same comparison in the mutants indeed suggest that bipolar cells degenerate and are lost. However, the number of amacrine cells at P10 is similar to that at P4. The authors conclude that there is also degeneration of amacrine cells but an alternative possibility is that cells are no longer generated and thus the population does not expand as in controls.

The HandE staining in Figure 1F seems to indicate the presence of cells in the ONL, presumably photoreceptor nuclei that however are strongly decreases at P30. The data in Figure 3 and the related supplementary information indicate a strong decrease of photoreceptor specific-gene expression. Again, the author conclusion is that NMNAT1 is required for postnatal survival of photoreceptors. Although it is clear that part of the phenotype is linked to the activation of cell death pathways, the authors should also consider the possibility that NMNAT1 may be also required for the generation of a proper number of photoreceptors. This needs to be determined. Markers such as Otx2 or Crx could be used to determine if the same number of photoreceptors are generated and then die perhaps because they cannot terminally differentiate or they are simply not generated in the same amount, as it may also happen for amacrine and even bipolar cells. This is particularly important because the Six3-Cre is an early driver and thus NMNAT1 absence could affect progenitor cells. Furthremore, the authors state "In conclusion, this study presents the most comprehensive evaluation of NMNAT1-associated retinal dysfunction to date and suggests crucial roles for nuclear NAD+ in the proper development and early survival of the mammalian retina". This conclusion at the moment is an overstatement as the development function of NMNAT1 is not addressed.

The idea that there might be an impact on progenitors is possibly supported by comparing the present results with those obtained by Eblimit et al., 2018 as well as those by Kuribayashi et al., 2018. In both cases knock down of NMNAT1 occurred with a different timing that correlated with less severe phenotypes.

The authors should also consider discussing their results with those of Eblimit et al., 2018 in more details, given that in the latter the reported use of photoreceptor-specific drivers suggests that reduction of the INL might be secondary to the loss of photoreceptors.

*Reviewer #2:*

NMNAT1 is known to be causal to Laber congenital amaurosis (LCA) which is congenital retinal dystrophies that appears at an early age. In this paper, the authors aimed to reveal the roles of Nmnat1 in retinal development by using retina specific conditional knockout mice. Interestingly, the loss of Nmnat1 by Six3-cre resulted in severe perturbation of the retinal development. The authors claimed the degeneration of the photoreceptors, but the results suggested lack of rod differentiation rather than degeneration since no evidence showing rod differentiation in the mice was included. In contrast, interneurons were once differentiated, then disappeared in the postnatal period. Since the special expression pattern of Cre or NMNAT1 in the retina was not available, it made difficult to relate the complex and cell type specific phenotype and Nmnat1. There are several other papers examining the role of Nmnat1 in retinal development in in vitro and ex vivo, precise comparison of the methods and data among the works may give interesting insight.

This paper describes examination of retinal phenotype of Nmnat1 by using retina specific knockout mice of Nmnat1. The paper contains the results of variety of analyses i.e. histological analysis, transcriptome analysis, and metabolic pathway examination. However, the whole part of this manuscript is descriptive, and did not give mechanisms how NMNAT1 regulates retinal differentiation. The authors claim that the severe photoreceptor degeneration was observed in this mice; however, I think that the phenotype showed lack of rod photoreceptor differentiation but not degeneration.

1. The Six3-cre mouse was used in this work, but this mouse line is known to show chimeric expression pattern of cre in the retina. i.e. "More detailed analysis of the recombination pattern of Six3-cre in the mature retina suggested it may incompletely excise target genes of interest. JNS 2007, 27, 12707-.". The authors observed strong phenotype in the central retina but not in the peripheral. Does it correlate with the expression pattern of the Cre protein? They should examine spatial expression of Cre or Nmnat1 in the CKO in retinas of several different developmental stages.

2. They did not perform any detailed examination of the retinal progenitor cells. Staining of progenitor markers and proliferation antigens of embryonic as well as P0 to P4 retinas may give useful information to understand the phenotype.

3. The authors claims that they observed photoreceptor degeneration, and NMNAT1 is crucial for the early survival of photoreceptors. If the authors observe the transition of rod photoreceptor that is once differentiation and then disappearance of them, this statement is appropriate. However, since neither Rhodopsin nor Recoverin expressed in the P4 retina, I think that failure of rod differentiation is suitable description to account for this phenotype.

4. RNA-seq data showed decreased cone related genes. The authors should confirm differentiation of cone in embryonic as well as postnatal retina of the CKO mice.

5. The retina at P4 showed no OPL, and previous literature indicated that the OPL is established by the interaction of photoreceptor and horizontal cells. Is horizontal cell differentiate in the CKO retina? In addition, how about Mueller glia differentiation in the CKO retina?

6. The authors observed lack of Synaptophysin, and I wondered postsynaptic formation of horizontal and bipolar cells.

7. Figure 1D lacks the scale bar, and that made us difficult to examine the panels.

6. Is it possible to rescue the phenotype of the CKO retina by administration of NAD or NMN locally or systemically?

*Reviewer #3:*

The authors aimed to disentangle the peculiar and complex aspects of NMNAT1-dependent retinal degeneration, with a particular focus on the underlying metabolic and transcriptional changes, used to hypothesize a mechanistic scenario explaining the pathophysiological picture associated with mutations at this locus. Overall, the paper is very well written and logical and provides a very comprehensive analysis spanning from detailed phenotypic analysis to metabolomics and transcriptomics approaches. This is definitely the major strength of the paper: experimental comprehensiveness and multilevel analysis. Using this approach, in general, the authors managed to provide interesting insights on the putative mechanistic scenario on the basis of the retinal degeneration caused by NAD+ and NMNAT1 deficiencies. Moreover, the cell-specific analysis provided also brings value and specificity to the study. Although the aim was obviously ambitious, the results provided well support most of their conclusions. In spite of this, I do have a major concern about the paper submitted. In general, the authors cite several papers with overlapping data/approaches to those proposed in the submitted manuscript (i.e. Lin et al., 2016, Greenwald et al., 2021, Wang et al., 2017, Lundt et al., 2021; Kuribayashi et al., 2018, Sasaki et al., 2020b). I understand that this is a hot topic in the field, that converging data is understandable, and that independent studies providing similar results are an extraordinary opportunity to validate the strength of the findings. However, I would encourage the authors to better state/define what differentiates some of their findings from those reported in the mentioned papers which used both retina or photoreceptor-specific NMNAT1 ablation, as well as metabolomic and transcriptional profiling of such transgenic animals. Although the authors use a very logical and comprehensive approach (and did a great job in presenting their findings with clarity), in several parts I kind of struggled a bit in recognizing the novelty of what they are bringing, and which kind of knowledge gaps they aim to fill, beyond the rhetoric of scientific writing.

A second aspect I would like to point out concerns the conclusions drawn by the authors concerning the different sensitivity of the distinct cell populations to the altered NAD+ homeostasis/NMNAT1 knockdown. It is well possible that amacrine, photoreceptor, bipolar cells, and RGCs respond differently to the tested conditions. However, it is also possible that photoreceptor cell degeneration (which occurs relatively early) represents also a cause of the later impairments in the other cell populations, affecting in turn their survival. Definitely, photoreceptors are, among the mentioned cells, those where altered REDOX chemistry (where NAD+ plays a crucial role) can have a bigger impact. I think that in the model proposed, it is not very clear why, the other cell types are affected as well, although later (even at P30 for RGCs). In other words, can we be sure that the degeneration observed in the different cell populations is actually cell-autonomous and not the result of a malfunctioning of one of the cell types? Are the effects on later cell types specific or just a consequence of the earlier impairment in photoreceptors? Maybe the authors could comment on this.

– Line 60: I found it interesting that retinas are particularly sensitive to NAD+ metabolism perturbations. Although the authors provide several examples in the literature, I would ask them to further comment (also in the paper) on why this tissue should be particularly reliant on NAD+ metabolism compared to the others (that to my knowledge should require a fine-tuned NAD+ homeostasis as well), also considering what they state in lines 71-75.

– Line 76: The authors define the other NMNAT as isoforms, but they are actually different genes, so I would define them as "other NMNAT genes", or, in case something is known about their phylogeny, use the more appropriate term paralogs (as I assume they are the results of vertebrate genome duplications).

– Line 123: what is the general range of knockout efficiency with the Cre-lox system, and what was the efficiency in the other papers that used a similar strategy to investigate NMNAT1 (see above). Can this explain some inconsistencies with some of the other studies?

– In Figure 1 legend it is not clear how many animals have been used as biological replicates (Figure 1C-J).

– Figure 1B: t-test is often accepted with 3 biological replicates, yet, theoretically, to assume a normal distribution you need more than 3 datapoints, so I would suggest a Mann-Whitney test (same applies to Figure 2, Figure 4E, J and supplementary figures using t-test, like Figure S3C, Figure S4E).

– Figure 1H-J: It is not possible in this case to use a t-test, as you would need anyway to correct for multiple comparisons even if you are interested in the KO vs wt comparison at a single specific distance. In this case, the authors should use an ANOVA or, given the consideration above (t-test vs Mann-Whitney test), with a non-parametric equivalent, like Kruskal-Wallis (in both cases corrected for multiple comparisons, as the more comparisons you are interested in, the more likely is to find something significant).

– Line 156: In this section, some % are given, yet it is not clear how many animals have been used to calculate such numbers. Did the authors use only one animal per timepoint? Even without statistical constraints (this is not the goal here), I would suggest at least 2-3 specimens.

– Line 332: not necroptosis as well? (See lines 218-221)

– As far as I see, no statistical analysis has been performed for Figure 5C, E, H. Is there a valuable and specific reason? If not, I would include a statistical significance assessment.

– Lines 445-452: I am not fully convinced by the explanation concerning the contrasting evidence between the manuscript and Kuribayashi et al., 2018. The current study reports defects in amacrine and bipolar cells in relatively later stages, so I am not sure whether an earlier knockdown can explain these discrepancies.

– Line 606: although I recognize this study as very well done and comprehensive, I do not think that the authors themselves should define their manuscript as the most… I suggest they change "the most… to date" with "a". I also would like to remember the authors that yes, their study might be the most comprehensive, but they largely base their study on a plethora of studies using very similar approaches and providing similar advancements in the field.

– Especially for scientists from different fields, I would spend few words explaining the relationships between the different cell types mentioned (how they are contributing to retinal development and vision, for instance).

---

## [Author Response]

Essential revisions:1) The authors need to experimentally address the potential role of NMNAT1 in retinal progenitor differentiation, with especial consideration of amacrine, photoreceptor and bipolar progenitors. The reviewers share the opinion that the data presented in the manuscript could be the result of defective specification/differentiation and not only linked to degeneration as the authors claim. Thus, staining for progenitor markers and proliferation antigens at embryonic and postnatal stages is required. Any other approach aimed at addressing this issue will be equally valid.

We thank the reviewers for raising this important distinction—in response, we have considerably expanded Figures 2 and 3 to include staining for several proliferation and progenitor markers in ages preceding and succeeding gross retinal degeneration in our model. Phosphorylated histone H3 (Figure 3—Supplement 1), a marker for M phase cells (Prigent and Dimitrov, 2003), Phosducin (Figure 3), a marker of developing cone photoreceptors (Rodgers et al., 2016), and OTX2, a regulator of photoreceptor and bipolar cell terminal differentiation (Beby and Lamonerie, 2013) (Figure 3) reveal severe defects in photoreceptor terminal differentiation but only modest reductions in progenitor proliferation in NMNAT1 knockout retinas. In addition to Phosducin, we examined cone specification in our model using anti-M-opsin and peanut agglutinin (PNA), a lectin which labels developing cone outer segments (Blanks and Johnson). At P4, this suggests cone specification defects equal in magnitude to rod specification defects (which we demonstrated by staining for rhodopsin and recoverin) in knockouts.

With regards to bipolar cell specification, several attempts at checking the expression of MASH1 a supposed bipolar differentiation marker (Hatakeyama and Kageyama, 2004) and PKC-α (a well-characterized marker of mature bipolar cells) were unsuccessful—the former due to antibody issues, and the latter since PKCa in our model is not detectable until later ages, at which point knockout retinas are nearly completely degenerated. While NMNAT1 may play a role in the terminal differentiation of retinal bipolar cells (supported perhaps by aforementioned OTX2 misexpression in knockouts), we note that CHX10 is involved in bipolar cell differentiation (Hatakeyama and Kageyama, 2004) The fact that we observe equal amounts of CHX10-positive cells in P4 knockout retinas—an age at which we report complete abrogation of M-opsin, rhodopsin, and recoverin expression—suggests that any potential effects of NMNAT1 loss on bipolar specification are at least less severe than those observed for photoreceptor terminal differentiation. We note also that levels of *Mash1* and its partner *Math3* are unchanged in our P4 RNA-sequencing dataset, and *Mash1* is similarly unchanged in E18.5 knockouts.

While we were unable to specifically probe amacrine cell specification, we have added staining for calbindin, which labels both horizontal and amacrine neurons (Wassle et al., 1998). At P4, quantification of calbindin-positive amacrine cells (identified by their location proximal to the inner plexiform layer) shows no significant change in amacrine cell numbers in knockouts, mirroring our Calretinin analyses. Classical and contemporary birth-dating studies (Cepko, 2014; Cherry et al., 2009; Voinescue et al., 2009) indicate that the vast majority of retinal amacrine cells area born well before P4—an age at which we report no change in calbindin or calretinin-positive amacrine cells in NMNAT1 knockouts. Thus, while we detect an increase in calretinin-positive amacrine cells in wild-type controls between P4 and P10, this observation likely reflects calretinin expression and not amacrine cell specification. For these reasons, we consider it unlikely that NMNAT1 influences amacrine cell differentiation; however, since we have not conclusively ruled out this possibility, we have included a brief discussion of this matter in the main text. A more exhaustive examination of amacrine cell specification in NMNAT1 knockout retinas presents a potential future avenue of study.

2) Analysis of cone, amacrine and Müller cell specification/differentiation should also be included as morphological and RNA-seq data suggests possible decrease of their specification/differentiation.

We thank the reviewers for the motivation to examine cone terminal differentiation using the aforementioned Phosducin and M-opsin/PNA stains, which revealed severe defects in cone specification in NMNAT1 knockout retinas. Müller glia are among the latest-born retinal cell types (Cepko 2014), and in our wild-type controls are only detectable on staining after ~P8, by which time there is considerable degeneration in knockout retinas. Thus, we do not consider our pan-retinal knockout model to be particularly informative for identifying a role for NMNAT1 in Müller glia development—perhaps a glia-specific NMNAT1 knockout would be better suited for this question.

We would like to stress that we do not claim that NMNAT1 functions only in the terminal differentiation of photoreceptors. Rather, we attempt to draw a distinction between the phenotypes we observe in photoreceptor cells (early transcriptional reduction in a cluster of visual transduction genes, complete abrogation of opsin and recoverin expression, rapid degeneration) versus non-photoreceptor cells (largely unchanged transcriptional signatures, persistence of cell-type markers, delayed degeneration). We use this distinction to (1.) demonstrate that NMNAT1 loss during retinal development affects cell types beyond photoreceptors, and (2.) propose an especially salient role for NMNAT1 in photoreceptor terminal differentiation, a role which has not to this point been described. We have revised the main text to make this intent clearer.

3) A better description of the knockout efficiency with the Cre-lox system used and its comparison with those used in the other studies to investigate NMNAT1 requirement in the retina is needed.

We have included a comparison of our knockout efficiency with that of (Sasaki et al., 2020a) in the main text. Unfortunately, we were unable to accurately determine the knockout efficiencies of the models presented in Wang et al., or Eblimit et al., —the former because the publication does not include this information and the latter because only a non-quantitative immunofluorescent image was provided to demonstrate NMNAT1 knockout. We do note, however, that the *Six3-Cre* mice used in our study have previously been validated by crossing with a transgenic Cre reporter line, which showed uniform *Six3-Cre* expression throughout the retina (Christiansen et al.,). However, the reviewers raise an important point considering potentially lower *Six3-Cre* expression in the peripheral retina—for this reason, we focus our analyses on the central retina and do not believe this possibility to influence the validity of our findings.

4) Statistical analysis of the data should be revised in Figure 1 and added in Figure 5.

We appreciate these comments and apologize for any confusion regarding statistics in Figure 5—we had intended the color coding of the bars to denote significance, opting against including asterisks above the bars for fear that they would be misinterpreted as data points. To make this color coding clearer, we have emphasized the color legend and added a brief comment to the figure legend. We also added asterisks to denote significance in the heatmap in Figure 5G. With regards to using non-parametric significance tests in Figure 1, we have replaced the analysis in Figure 1B with a Mann-Whitney U test and updated the legend accordingly. However, we have opted to keep the multiple t-test analysis in panels H-J because we only intended to compare thicknesses between control and knockout retinas at each individual distance, and we feel that an assumption of normality among retinal thickness measurements is not unreasonable in this instance. While we are careful to correct for multiple comparisons in our large-scale transcriptomics dataset, any detrimental effects of making 8 comparisons versus 1 are minimal, and debate exists as to the overall appropriateness of multiple comparison correction in every case (Althouse 2016). We include all exact p-values in our source data file and have added 95% confidence intervals of these thickness measurements to the main text—importantly, we observe identical trends in retinal thickness throughout many independent immunostaining experiments later in the paper, so we believe that the specific statistical test used in these panels does not influence the conclusions to an appreciable degree.

5) A number of different studies have used similar approaches (retina-specific NMNAT1 KO or KD, transcriptomics, and metabolomics) to shed light on NMNAT1 function in the retina. The authors need to clarify what are the knowledge gaps they are trying to fill in and should discuss better their findings in relation to the existing data (i.e. in relation to those of Eblimit et al., 2018 and Kuribayashi et al., 2018).

To address this concern, we have added several paragraphs in the main text which (1.) better explain any potential inconsistencies between our study and that of Kuribayashi et al., (2.) discuss the possibility of cell-autonomous roles of NMNAT1 in relation to findings of Wang et al., and (3.) better clarify the novelty of our findings. With full respect to the authors of these previous studies, we would like to point out that the mere existence of these publications overestimates the extent to which NMNAT1 knockout retinas have been characterized—Wang et al., provide only a 7-page (omitting references) preprint with a single figure, and Eblimit et al., focus mostly on a disease-linked NMNAT1 mutation (which they demonstrate does not reduce NMNAT1 expression), providing only histological analyses of their various NMNAT1 conditional knockout models. We are grateful to extend some of the findings of Kuribayashi et al., to an in vivo setting, but our in vivo cell-type marker analyses, transcriptomics-guided investigation of cell death and photoreceptor differentiation, and comprehensive metabolomics analyses have not previously been reported in an NMNAT1-deficient animal model.

6) The authors may also consider rescuing the phenotype of the CKO retina by the administration of NAD or NMN, if experimentally feasible.

While we appreciate this suggestion and see the potential value of this experiment, uncertainties concerning the timing and proper delivery of NAD to newborn pup retinas argues against the current feasibility of this approach (NMN, meanwhile, is an NMNAT1 substrate and is considerably elevated in our knockout retinas (Figure 5C)).

In addition to the above changes, we note the following changes in response to individual reviewer comments:

– Scale bars have been added to figure 1

– We have changed references of NMNAT1 “isoforms” to “paralogs” or “enzymes” (thank you for catching this)

– We have added information to all figure legends indicating the number of biological replicates used for each experiment

– We have changed “the most comprehensive..” to “a comprehensive…” at the end of the Discussion section, and added more details to better contextualize our findings

References

Althouse, A. D. (2016). Adjust for Multiple Comparisons? It’s Not That Simple. The Annals of Thoracic Surgery, 101(5), 1644–1645. https://doi.org/10.1016/j.athoracsur.2015.11.024

Beby, F., and Lamonerie, T. (2013). The homeobox gene Otx2 in development and disease. Experimental Eye Research, 111, 9–16. https://doi.org/10.1016/j.exer.2013.03.007

Blanks, J. C., and Johnson, L. V. (n.d.). Specific Binding of Peanut Lectin to a Class of Retinal Photoreceptor Cells. 25, 12.

Cepko, C. (2014). Intrinsically different retinal progenitor cells produce specific types of progeny. Nature Reviews Neuroscience, 15(9), 615–627. https://doi.org/10.1038/nrn3767

Cherry, T. J., Trimarchi, J. M., Stadler, M. B., and Cepko, C. L. (2009). Development and diversification of retinal amacrine interneurons at single cell resolution. Proceedings of the National Academy of Sciences, 106(23), 9495–9500. https://doi.org/10.1073/pnas.0903264106

Christiansen, J. R., Kolandaivelu, S., Bergo, M. O., and Ramamurthy, V. (n.d.). RAS-converting enzyme 1-mediated endoproteolysis is required for trafﬁcking of rod phosphodiesterase 6 to photoreceptor outer segments. 5.

Di Stefano, M., Nascimento-Ferreira, I., Orsomando, G., Mori, V., Gilley, J., Brown, R., Janeckova, L., Vargas, M. E., Worrell, L. A., Loreto, A., Tickle, J., Patrick, J., Webster, J. R. M., Marangoni, M., Carpi, F. M., Pucciarelli, S., Rossi, F., Meng, W., Sagasti, A., … Conforti, L. (2015). A rise in NAD precursor nicotinamide mononucleotide (NMN) after injury promotes axon degeneration. Cell Death and Differentiation, 22(5), 731–742. https://doi.org/10.1038/cdd.2014.164

Hatakeyama, J., and Kageyama, R. (2004). Retinal cell fate determination and bHLH factors. Seminars in Cell and Developmental Biology, 15(1), 83–89. https://doi.org/10.1016/j.semcdb.2003.09.005

Prigent, C., and Dimitrov, S. (2003). Phosphorylation of serine 10 in histone H3, what for? Journal of Cell Science, 116(18), 3677–3685. https://doi.org/10.1242/jcs.00735

Rodgers, H. M., Belcastro, M., Sokolov, M., and Mathers, P. H. (2016). Embryonic markers of cone differentiation. Molecular Vision, 22, 1455–1467.

Voinescu, P. E., Kay, J. N., and Sanes, J. R. (2009). Birthdays of retinal amacrine cell subtypes are systematically related to their molecular identity and soma position. The Journal of Comparative Neurology, 517(5), 737–750. https://doi.org/10.1002/cne.22200

Wässle, H., Peichl, L., Airaksinen, M. S., and Meyer, M. (1998). Calcium-binding proteins in the retina of a calbindin-null mutant mouse. Cell and Tissue Research, 292(2), 211–218. https://doi.org/10.1007/s004410051052